# COMMUNICATION EFFICIENT LLM PRE-TRAINING WITH SPARSELOCO

## ABSTRACT

Communication-efficient distributed training algorithms have received considerable interest recently due to their benefits for training Large Language Models (LLMs) in bandwidth-constrained settings, such as across datacenters and over the internet. Despite reducing communication frequency, these methods still typically require communicating a full copy of the model's gradients—resulting in a communication bottleneck even for cross-datacenter links. Furthermore, they can slightly degrade performance compared to a naive AdamW DDP baseline. While quantization is often applied to reduce the pseudo-gradient's size, in the context of LLM pre-training, existing approaches have been unable to additionally leverage sparsification and have obtained limited quantization. In this work, we introduce SparseLoCo, a communication-efficient training algorithm for LLMs that effectively leverages error feedback with TOP-$k$ sparsification and 2-bit quantization to reach extreme sparsity as low as 1–3% while outperforming full-precision DiLoCo. Our key observations are that outer momentum can be locally approximated by an error feedback accumulator combined with aggressive sparsity, and that sparse aggregation can actually improve model performance. We empirically demonstrate in a range of communication-constrained LLM training settings that SparseLoCo provides significant benefits in both performance and communication cost.

## 1 INTRODUCTION

Frontier language models pre-trained on internet-scale data have led to considerable breakthroughs in recent years. However, due to their growing parameter counts, effectively training these models across expensive datacenter hardware while retaining efficiency—a central goal due to the resources spent on these runs—is becoming increasingly challenging. On the other hand, due to the increasing availability of globally distributed computational infrastructure across the world, the pre-training of large-scale models over the internet has recently garnered increasing interest (Jaghouar et al., 2024). Similar to training over the internet, pre-training across multiple datacenters requires mitigating the large communication overhead incurred by aggregating updates between workers.

In the context of LLM pre-training, several approaches have been proposed to reduce data-parallel communication cost. Among them are DiLoCo (Douillard et al., 2023b), a variant of LocalSGD (Stich, 2018; Reddi et al., 2020), as well as methods compressing communicated tensors and leveraging error feedback (Peng et al., 2024; Ahn & Xu, 2025; Wang et al., 2023) to mitigate information loss. These techniques have complementary advantages of (1) reducing the communication frequency and (2) reducing the size of communicated messages. Combining the two is potentially advantageous for bandwidth-constrained settings like training over the internet or across datacenters. However, existing works focused on LLM pre-training using DiLoCo (Charles et al., 2025) do not take full advantage of compression schemes.

Indeed, combining these approaches raises the challenge of how to incorporate error feedback with an outer momentum, which is known to be important for DiLoCo's performance. Our key observation is that when aggressive TOP-$k$ sparsification is combined with error feedback on DiLoCo pseudo-gradients, two effects emerge: (a) error feedback naturally acts as a local approximation of outer momentum, and (b) sparse aggregation is induced on the pseudo-gradients, a property recently shown in model merging contexts to improve performance (Yadav et al., 2023; Davari & Belilovsky,

2024). Building on this, we introduce **SparseLoCo**, which replaces global outer momentum with a single error feedback accumulator, thereby unifying infrequent and sparsified communication. This enables aggressive TOP-$k$ sparsification and quantization of pseudo-gradients, while outperforming full-communication DiLoCo and frequency-compressed baselines.

Our contributions can be summarized as follows:

- We demonstrate that DiLoCo's outer momentum can be replaced with a local momentum, which we link to TOP-$k$ with error feedback on pseudo-gradients.
- Leveraging this observation, we introduce SparseLoCo, a novel algorithm that blends the benefits of multi-iteration methods like DiLoCo with TOP-$k$ sparsification and error feedback without compromising on performance or communication cost.
- Through our extensive experiments, we demonstrate that SparseLoCo can significantly reduce the communication volume compared to existing LLM training methods (e.g., DiLoCo and DeMo), while simultaneously outperforming them.

## 2 RELATED WORK

**Federated Learning** In the federated learning literature, communication efficiency has been a central focus from the outset, as participating clients often operate over highly constrained and heterogeneous networks. A canonical example is Federated Averaging (FedAvg) (Konečný et al., 2016; McMahan et al., 2017), which reduces communication frequency by performing multiple local updates before averaging model parameters. Other works explore compressed updates through sketching or quantization in the context of federated learning (Rothchild et al., 2020; Reisizadeh et al., 2020). Beyond reducing communication overhead, numerous approaches such as SCAF-FOLD (Karimireddy et al., 2020) and FedProx (Li et al., 2020) address the unique challenge of data heterogeneity—where each client's dataset may follow a different distribution—by introducing control variates or proximal terms to stabilize convergence. Related to our work Mitchell et al. (2022) consider pseudo-gradient compression in the FL setting. While our work shares federated learning's emphasis on reducing communication overhead, it differs fundamentally in scope: we focus on large-scale pre-training of LLMs in settings with homogeneous data partitions (e.g., sharded web-scale corpora), where heterogeneity-mitigation strategies are unnecessary while achieving performance that can match standard data parallel schemes at equivalent FLOPs is the paramount Douillard et al. (2023a).

**LocalSGD and extensions to LLM training** Local Stochastic Gradient Descent (LocalSGD) (Stich, 2018) is a widely studied approach for reducing communication in distributed training by allowing workers to perform multiple local updates before synchronizing. Stich (2018) formally introduced the method and proved its convergence, while Lin et al. (2018) highlighted that LocalSGD can lead to improved generalization compared to simply increasing the batch size. Extensions of LocalSGD include SlowMo (Wang et al., 2019), which incorporates a slow outer momentum to stabilize training in datacenter-style environments—while still using SGD as the inner optimizer—and meta-learning approaches (Joseph et al., 2025) that adapt the aggregation function for improved performance. However, these approaches were not shown to scale well to pre-training in Ortiz et al. (2021). More recently, DiLoCo (Douillard et al., 2023b) adapted the LocalSGD framework to Large Language Model (LLM) pre-training, demonstrating that replacing the inner optimizer with AdamW and a nesterov momentum outer optimizer can yield substantial benefits. Our work builds upon this line of research by enabling aggressive TOP-$k$ sparsification of the communicated pseudo-gradients in a LocalSGD-style framework, something that prior methods have not achieved while maintaining or improving upon state-of-the-art LLM training performance. Finally, Douillard et al. (2025) and Fournier et al. (2024) consider communicating a small subset of model parameters more frequently instead of communicating a message the size of the model infrequently by allowing the models to desynchronize while still remaining relatively close to each other (in terms of, e.g., consensus distance). SparseLoCo, on the other hand, maintains the benefit of infrequent communication while communicating a small-sized message and maintaining model synchronization.

**Error Feedback and Compressed Updates** Error feedback (EF) has been extensively studied, particularly from a theoretical perspective, as a means to compensate for the information loss introduced by various gradient compression methods (Seide et al., 2014; Karimireddy et al., 2019; Stich

& Karimireddy, 2019). It has been combined with various compression techniques, including quantization, sparsification (Shi et al., 2019), and low-rank approximation in (Vogels et al., 2019; Ahn & Xu, 2025). In the single local step setting it has been applied to LLMs in recent works (Wang et al., 2023; Peng et al., 2024; Zhao et al.). EF21-SGDM (Fatkhullin et al., 2023) analyzed how to combine error feedback with momentum, introducing a momentum-compatible variant that requires two accumulators and is largely focused on theoretical aspects and does not address the multi-iteration setting or the practical challenges of LLM pre-training. QSparseLocalSGD (Basu et al., 2019) is, to our knowledge, one of the few works that combines multi-iteration methods such as LocalSGD with error feedback, but its focus was on theoretical analysis with non-adaptive optimizers and without outer momentum which is crucial to high performance in the LLM setting. In contrast, our work targets the LLM pre-training regime and develops a method to combine aggressive TOP-$k$ compression and error feedback with an efficient approximation of outer momentum. DeMo (Peng et al., 2024) considers EF with DCT encoding and TOP-$k$ compression in the LLM setting, demonstrating it can achieve competitive performance, but without incorporating local updates or the ability to leverage adaptive optimizers. Similarly, CocktailSGD (Wang et al., 2023) uses error feedback with multiple compression operators in an LLM fine-tuning setting, yet does not explore the integration of local iteration methods. Our work studies the combination of these approaches in the context of LLMs and more generally in the context of modern variations of multi-iteration methods that have been shown to scale to pre-training.

## 3 METHODOLOGY

In this section, we first review DiLoCo. We then propose replacing the global outer momentum in DiLoCo with per-replica local outer momentum (LOM), where each replica maintains its own accumulator, an approach that will be used to empirically analyze the need for global momentum. Finally, we present our proposed method, **SparseLoCo**, which combines TOP-$k$ compression with DiLoCo's infrequent communication.

### 3.1 BACKGROUND AND NOTATION

Consider the DiLoCo/FedOpt (Douillard et al., 2023a; Reddi et al., 2020) framework, which utilizes the following basic rule on each worker or replica to produce a pseudo-gradient at each outer step, $\Delta_r^{(t)}$, as follows:

$$\theta_r^{(t)} \leftarrow \text{InnerOpt}_H\left(\theta^{(t-1)}; \mathcal{D}_r\right), \quad \forall r \in [R],$$
$$\Delta_r^{(t)} \leftarrow \theta^{(t-1)} - \theta_r^{(t)}.$$

Here, $H$ corresponds to the number of inner steps of the optimizer (typically AdamW), and $R$ is the number of replicas. DiLoCo, which corresponds to an instantiation of FedOpt with AdamW as the inner optimizer and outer (server) momentum using Nesterov (Dozat, 2016), is given as follows:

$$\bar{\Delta}^{(t)} \leftarrow \frac{1}{R}\sum_{r=1}^{R}\Delta_r^{(t)},$$
$$m^{(t)} \leftarrow \beta\, m^{(t-1)} + \bar{\Delta}^{(t)}, \quad \tilde{\Delta}^{(t)} \leftarrow \bar{\Delta}^{(t)} + \beta\, m^{(t)},$$
$$\theta^{(t)} \leftarrow \theta^{(t-1)} - \alpha\, \tilde{\Delta}^{(t)}.$$

### 3.2 LOCAL OUTER MOMENTUM

We first propose a variant of DiLoCo that utilizes a per-replica local outer momentum instead of the unified global momentum. The goal of this algorithm is to provide insight into how well the outer momentum can be locally approximated. We denote this algorithm DiLoCo-LOM (Local Outer

Momentum):

$$m_r^{(t)} \leftarrow \beta\, m_r^{(t-1)} + \Delta_r^{(t)}, \quad \tilde{\Delta}_r^{(t)} \leftarrow \Delta_r^{(t)} + \beta\, m_r^{(t)},$$

$$\tilde{\Delta}^{(t)} \leftarrow \frac{1}{R} \sum_{r=1}^{R} \tilde{\Delta}_r^{(t)},$$

$$\theta^{(t)} \leftarrow \theta^{(t-1)} - \alpha\, \tilde{\Delta}^{(t)}.$$

Here, the outer momentum is updated locally, solely based on the local pseudo-gradient, while the final update is based on the average of the local momentum accumulators $\tilde{\Delta}^{(t)}$. Note that typical implementations of DiLoCo store the outer momentum locally on each replica, meaning that DiLoCo-LOM does not add any memory overhead compared to the global momentum variant. We show that the DiLoCo-LOM update exactly matches the DiLoCo update in Appendix J.

Building up to SparseLoCo, we consider an additional method, denoted DiLoCo-LOM-Sub-$k$, where the local momenta have their largest components removed at the end of each outer step:

$$m_r^{(t)} \leftarrow m_r^{(t)} - \text{TOP-}k\left(m_r^{(t)}\right)$$

This allows us to study the impact of TOP-$k$ subtraction, used in error feedback, without sparsifying the pseudo-gradient.

### 3.3 SPARSELOCO: SPARSE AGGREGATION MEETS LOCAL OUTER MOMENTUM

We now introduce SparseLoCo, which blends TOP-$k$ sparsification and error feedback in place of the local outer momentum. We consider error feedback, $e_r$, applied to the pseudo-gradients, which we denote as OuterEF:

$$e_r^{(t)} \leftarrow \beta\, e_r^{(t)} + \Delta_r^{(t)}$$

$$\hat{\Delta}_r^{(t)} \leftarrow Q\left(\text{TOP-}k\left(e_r^{(t)}\right)\right), \quad e_r^{(t+1)} \leftarrow e_r^{(t)} - \hat{\Delta}_r^{(t)}$$

$$\Delta^{(t)} \leftarrow \frac{1}{R} \sum_{r=1}^{R} \hat{\Delta}_r^{(t)},$$

$$\theta^{(t)} \leftarrow \theta^{(t-1)} - \alpha\, \Delta^{(t)}.$$

Here, $Q$ is the quantization function which allows further compression of the selected values. When $k$ is sufficiently small, OuterEF closely approximates the local outer momentum in LOM, since only a few components will be subtracted from $e_r$. On the other hand, unlike LOM and LOM-Sub-$k$, SparseLoCo only aggregates quantized sparse vectors, drastically reducing the message size needed for communication. Throughout the paper, we refer to the *communication density* as the fraction of coordinates in the pseudo-gradient that are transmitted at each outer synchronization step; for brevity, we simply write "density" referring to the same quantity. The full algorithm for SparseLoCo is given in Algorithm 1.

SparseLoCo uses a chunk-wise variant of the TOP-$k$ operation inspired by Xu et al. (2021); Peng et al. (2024). To do so, we first partition each 2D parameter tensor (e.g., attention and MLP weight matrices) into non-overlapping $64 \times 64$ blocks and each 1D tensor (e.g., layer-norm parameters) into contiguous chunks of size 4096, and then apply TOP-$k$ independently within each chunk. This has three benefits compared to applying it at the full-tensor or global level: (a) the cost of naively storing indices for transmission is significantly reduced as each chunk's index space is bounded. (b) TOP-$k$ applied to entire models or individual tensors can overemphasize correlated variables; thus, chunking can have benefits on performance as further discussed in Appendix B. (c) Finally, chunking can allow for more easily integrating tensor parallelism and FSDP, which often require sharding across tensors, thereby creating inefficiencies for TOP-$k$ operations over entire tensors or models.

## 4 EXPERIMENTS

Our experiments use 178M-, 512M-, and 2B-parameter LLaMA-style decoder-only transformer on DCLM (Li et al., 2024) using the LLaMA-2 tokenizer (Touvron et al., 2023). Following Hoffmann

---

**Algorithm 1** SparseLoCo

---

**Require:** initial parameters $\{\theta_r^{(0)}\}$, inner steps $H$, outer steps $T$, outer learning rate $\alpha$, error momentum $\beta$, workers $R$, per worker training data $D_r$.

1: **for** $t \leftarrow 1$ **to** $T$ **do**
2:     **for** $r \leftarrow 1$ **to** $R$ **do**

**Local inner loops**

3:         $\theta_r^{(t)} \leftarrow \theta_r^{(t-1)}$
4:         **for** $h \leftarrow 1$ **to** $H$ **do**          ▷ Local inner loops
5:             Sample $x \sim \mathcal{D}_r$
6:             $L \leftarrow f(x, \theta_r^{(t)})$
7:             $\theta_r^{(t)} \leftarrow \mathrm{AdamW}(\theta_r^{(t)}, \nabla L)$
8:         **end for**
9:         $\Delta_r^{(t)} \leftarrow \theta_r^{(t-1)} - \theta_r^{(t)}$          ▷ Pseudo-gradient

**Compression + Error Feedback**

10:         $e_r^{(t)} \leftarrow \beta\, e_r^{(t)} + \Delta_r^{(t)}$
11:         $\hat{\Delta}_r^{(t)} \leftarrow Q(\mathrm{TOP}\text{-k}(e_r^{(t)}))$          ▷ Transmit $\hat{\Delta}_r^{(t)}$
12:         $e_r^{(t+1)} \leftarrow e_r^{(t)} - \hat{\Delta}_r^{(t)}$

**Aggregate + Outer Update**

13:         $\Delta^{(t)} \leftarrow \frac{1}{R} \sum_{r=1}^{R} \hat{\Delta}_r^{(t)}$
14:         $\theta_r^{(t+1)} \leftarrow \theta_r^{(t)} - \alpha\, \Delta^{(t)}$
15:     **end for**
16: **end for**

---

et al. (2022), we allocate a token budget equal to $20\times$ the model size. Our experimental protocol follows (Charles et al., 2025). Unless otherwise stated, our main results are reported on the 512M-parameter model with $R{=}8$ workers, per-worker batch size $B{=}256$, and sequence length $L{=}2048$, yielding a global batch of $B \times L \times R \approx 4.19$M tokens per step. For SparseLoCo, we employ 2-bit quantization with a chunk size of 4096 (Non-overlapping square $64 \times 64$ grids for 2D parameters). We apply a short error feedback (OuterEF) freeze, where the error feedback $e_r$ is not utilized for the first 5% of the outer steps to improve training stability and performance (ablated in Table 11 in Appendix E). We further study scaling across model sizes (178M; Appendix F, 2B; Table 5), and number of workers $R \in \{16, 32\}$ in Table 4, and Appendix Tables 12, and 13. We report the hyperparameter sweep ranges, architectural details, and selected configurations in Appendix H. As baselines, we include DiLoCo and DeMo—two strong communication-efficient methods (one using local iterations and the other using error feedback) for LLMs—as well as a DDP AdamW baseline.

### 4.1 BUILDING INTUITION WITH LOCAL OUTER MOMENTUM

We first use the DiLoCo-LOM and DiLoCo-LOM-Sub-$k$ algorithms to empirically link the standard outer momentum in DiLoCo to the error feedback mechanism in SparseLoCo. In DiLoCo-LOM, each replica maintains a local outer momentum accumulator that is averaged only at synchronization. In DiLoCo-LOM-Sub-$k$, we subtract the largest entries of the local momentum after each synchronization, isolating the effect of TOP-$k$ subtraction while keeping the communicated pseudo-gradients dense. As

Table 1: **DiLoCo's global outer momentum is well approximated by Local outer momentum.**

| Method | Loss |
|---|---|
| DiLoCo | 2.760 |
| DiLoCo w.o. outer momentum | 2.868 |
| DiLoCo-LOM | 2.759 |
| DiLoCo-LOM-Sub-$k$ - 25% | 2.761 |

shown in Table 1, DiLoCo-LOM matches DiLoCo's loss—consistent with Proposition 1 (Appendix J)—and pruning 25% of the largest accumulator entries has a negligible impact, whereas removing outer momentum entirely degrades performance.

To quantify how closely local outer momentum tracks the target global outer momentum in DiLoCo, we maintain a reference global accumulator and, over the first 20 outer steps, compute the cosine similarity between this reference and each replica's local accumulator at corresponding steps. The average similarity between individual local accumulators and the global reference accumulator is $\geq 0.75$ for DiLoCo-LOM-Sub-$k$ (25%), indicating that removing the top components from local accumulators each outer step still allows them to remain a strong directional proxy for the global momentum and supporting our interpretation of SparseLoCo's error feedback state as a local approximation of DiLoCo's outer momentum.

Table 2: SparseLoCo compared to DiLoCo and gradient compression (DeMo). We show the size of the pseudo-gradients sent, the number of synchronizations, quantization supported, and loss. SparseLoCo outperforms other communication-efficient baselines in both communication efficiency and loss. All results are reported for 512M models pre-trained on a 10B-token budget with $R=8$ replicas. Here, *Density* denotes the communication density as the percentage of coordinates in the pseudo-gradient vectors that are non-zero (and therefore transmitter) at each synchronization step.

| Method | Density | Loss | Pseudo-Grad Size | # of Syncs | Quantization |
|---|---|---|---|---|---|
| AdamW DDP | 100% | 2.69 | 1.02 GB | 2445 | 16-bit |
| DiLoCo (H=15) | 100% | 2.76 | 512.40 MB | 163 | 8-bit |
| DeMo | 0.78% | 2.83 | 10.01 MB | 2445 | 8-bit |
| DeMo | 3.12% | 2.86 | 40.03 MB | 2445 | 8-bit |
| SparseLoCo (H=15) | 0.78% | 2.79 | 4.25 MB | 163 | 2-bit |
| SparseLoCo (H=15) | 3.12% | **2.70** | 17.01 MB | 163 | 2-bit |

## 4.2 SPARSELOCO

Table 2 compares SparseLoCo at $H=15$ against existing methods. We utilize 2-bit quantization for SparseLoCo with no observed loss degradation, while using the prescribed quantization settings for baselines (Douillard et al., 2023a; Peng et al., 2024). We observe that SparseLoCo obtains lower final loss than DiLoCo and DeMo baselines, while enjoying the simultaneous communication benefits of aggressively sparsified pseudo-gradients and reduced synchronization frequency. As SparseLoCo

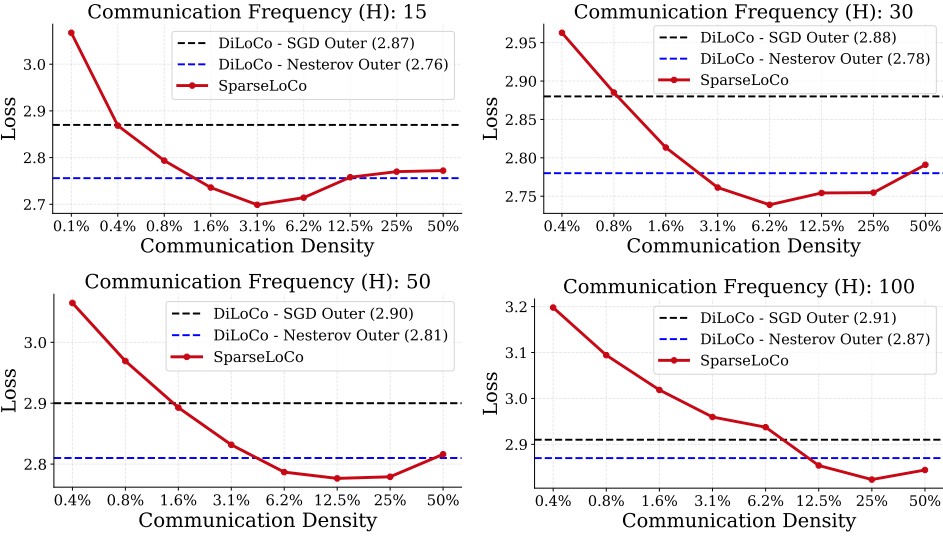

Figure 1: **SparseLoCo outperforms DiLoCo for** $H \in \{15, 30, 50, 100\}$ **communication intervals.** We evaluate SparseLoCo, DiLoCo, and DiLoCo without Nesterov for different communication intervals and at different sparsity levels for SparseLoCo. We report the best performance in each case. Crucially, SparseLoCo can outperform DiLoCo while communicating significantly less. We also observe that the optimal density grows with higher communication intervals. All experiments were conducted with $R = 8$ workers and 512M model size.

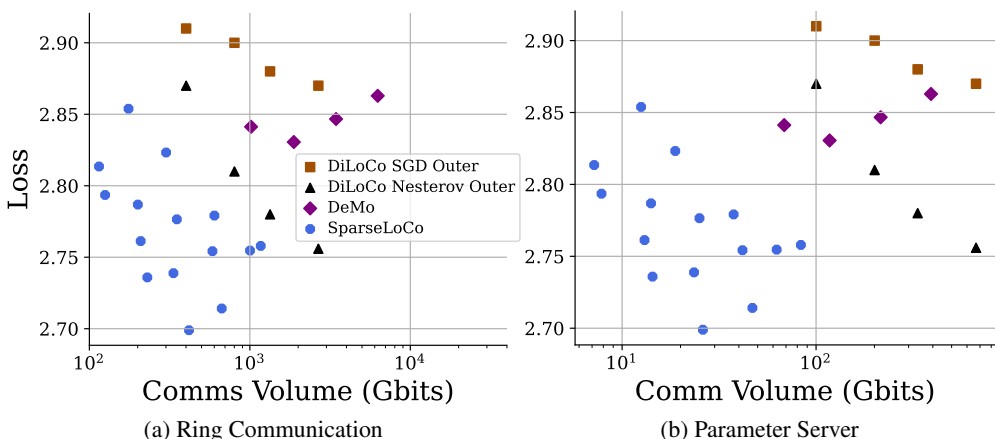

(a) Ring Communication  (b) Parameter Server

Figure 2: **SparseLoCo lies on the Pareto frontier between loss and communication volume.** We report communication volume (outbound) for two settings (A) ring communication topology (ring all-gather for SparseLoCo and DeMo, ring all-reduce for DiLoCo) (B) Parameter server. The points consider different $H$ for DiLoCo, different densities for DeMo, and combinations of both for SparseLoCo using 512M models. We observe that, in both cases, SparseLoCo is at the Pareto frontier.

inherently utilizes error feedback, SparseLoCo further reduces communication size by quantizing the sparsified values. We further compare the performance on simple downstream tasks relevant at this model scale in Table 3, demonstrating that the performance improvements are consistent.

**SparseLoCo performance at different sparsity levels and communication intervals** In Figure 1, we further demonstrate the performance improvements of SparseLoCo across TOP-$k$ densities and increasing $H \in \{15, 30, 50, 100\}$ values compared to well-tuned DiLoCo and DiLoCo without outer momentum baselines. We observe a trend that aligns with the hypothesis that SparseLoCo's OuterEF can provide similar benefits to DiLoCo's outer momentum. In particular, we first observe that not using outer momentum in DiLoCo leads to significant performance degradation and that this setting corresponds exactly to the fully dense case for TOP-$k$ (e.g., $k = 100\%$). With SparseLoCo, we observe that (i) extreme sparsity levels (when nearly nothing is sent, e.g., $0.05\%$) degrade performance. (ii) With increasing density (while remaining sparse), performance improves and eventually exceeds DiLoCo for all values of $H$. At this density levels, the EF buffer remains relatively dense and accumulates residual gradients, resembling a sparsified outer momentum in accumulating gradients (iii) Finally, as $k$ approaches dense communication, the EF buffer becomes more sparse (due to line 12 in Algorithm 1), trending towards the performance of DiLoCo with no outer momentum and again degrading performance. The same three regimes appear across all inner steps $H$. Furthermore, in Figure 4, we observe that this phenomenon happens with DeMo (Peng et al., 2024) as well, while the overall performance being inferior to SparseLoCo.

**SparseLoCo can outperform DiLoCo** Across all settings of the inner steps $H$, we observe regimes where SparseLoCo outperforms DiLoCo. A plausible explanation is that sparse aggregation at a well-chosen $k$ emphasizes high-saliency components and reduces interference among updates, echoing intuitions from recent model-merging work in multi-task fine-tuning (Yadav et al., 2023; Davari & Belilovsky, 2024).

**Higher sparsity is needed with fewer inner steps** We observe through Figure 1 a systematic pattern that the optimal value of SparseLoCo is reached at a higher sparsity level with fewer inner steps. This is consistent with the fact that higher inner steps communicate information from a larger total number of samples. Indeed, we would expect that a trajectory with more steps would have a larger support.

**SparseLoCo is at the Pareto frontier in communication volume** In Figure 2 we compare the communication volume of SparseLoCo to DiLoCo, DiLoCo w.o. outer momentum, and DeMo. The exact communication setting and the underlying implementation of the aggregation can have

Table 3: Benchmark (0-shot) accuracy (%; higher is better), **Best** is bold. We evaluate the same 512M pretrained models as in Table 2, using the 3.12% and 0.78% communication densities for SparseLoCo and DeMo, respectively, which correspond to the best performing configruations for each method in that table. We observe that SparseLoCo outperforms the DeMo and DiLoCo baselines across all benchmarks.

| Method | ARC-Easy | HellaSwag | PIQA |
|---|---|---|---|
| AdamW DDP | 44.99% | 36.08% | 65.34% |
| DiLoCo | 44.28% | 34.50% | 64.96% |
| DeMo | 41.92% | 32.37% | 64.09% |
| SparseLoCo | **45.24%** | **36.49%** | **65.23%** |

a significant impact on the communication volume. We consider two common setups from the literature—methods utilize either ring all-reduce or ring all-gather (Fig. A), or a parameter server (Fig. B). We observe that in both cases, SparseLoCo lies on the Pareto frontier while other methods have a strictly worse trade-off. We note that the results in Fig. A assume aggregation using a naive all-gather operation for implementation, while there is further potential to exploit the structure of the problem, for example, by summing overlapping indices along steps in the all-gather ring or utilizing specially designed all-reduce (Li & Hoefler, 2022). In Section A of the Appendix, we also discuss the communication measured during a live deployment of collaborative learning over the internet using SparseLoCo.

**SparseLoCo can be used with Ring All-Reduce as a drop-in for DiLoCo**   Although our analysis and motivation in the work focuses on aggressively compressing the per-iteration message size we note that in communication settings where efficient all-reduce is already available and preferred, SparseLoCo still provides significant benefit over DiLoCo while incurring no additional memory or compute overhead. Concretely, the aggregation step in Algorithm 1 Line 13 can be performed directly by an all-reduce over a sparse vector. This has two significant benefits over DiLoCo with all-reduce (AR): (1) As observed in Table 9 and Figure 1, the performance when $k$ is optimally selected is improved over DiLoCo and (2) the Outer error feedback naturally supports more aggressive quantization than the naive DiLoCo, allowing for 2-bit quantization to be used *without an additional accumulator*, unlike Thérien et al. (2025).

**SparseLoCo scales across model sizes, communication intervals, and number of replicas**   In Table 4, we evaluate scaling of DiLoCo and SparseLoCo with number of workers $R \in \{8, 16, 32\}$ using a 512M-parameter model scale and communication interval $H{=}50$. Scaling beyond 8 replicas without significant degradation is a known challenge Charles et al. (2025). We observe that SparseLoCo consistently outperforms DiLoCo with higher number of workers across all settings and across a number of densities, showing that it can help address the challenge of scaling the number of replicas. We also observe that with higher number of workers a lower density can sometimes be supported. In the Appendix 12 we also study the impact of the number of replicas at 178M model size. We also evaluate SparseLoCo in the highest communication intervals ($H{=}250$), for this we follow Charles et al. (2025) using an overtraining regime with a doubled token budget, where we see again SparseLoCo is able to achieve competitive performance with DiLoCo while improving communication. Finally, we run a larger scale model of size 2B-parameter model scale with $R{=}16$ workers and communication interval $H{=}50$, where SparseLoCo with $6.25\%$ density outperforms DiLoCo (Table 5). We can see that the benefits of SparseLoCo are maintained at this scale.

Table 4: The final evaluation loss of scaling number of replicas $R \in \{8, 16, 32\}$ for a 512M-parameter model with communication interval $H=50$ under different communication densities. SparseLoCo consistently outperforms DiLoCo as R increases. Best and second best results are presented in **bold**.

| Method | Density | Loss (R=8) | Loss (R=16) | Loss (R=32) |
|---|---|---|---|---|
| AdamW | 100.00% | 2.69 | 2.69 | 2.69 |
| DiLoCo | 100.00% | 2.81 | 2.87 | 2.93 |
| SparseLoCo | 0.78% | 2.97 | 3.00 | 3.09 |
| | 1.56% | 2.89 | 2.92 | 3.00 |
| | 3.12% | 2.83 | 2.86 | 2.92 |
| | 6.25% | 2.79 | 2.82 | **2.88** |
| | 12.50% | **2.78** | **2.80** | 2.91 |
| | 25.00% | **2.78** | 2.84 | 3.02 |

Table 5: Evaluation loss and benchmark (0-shot) accuracy of 2B-parameter LLMs with $R=16$ contributing peers. Best in **bold**.

| Method | Val Loss | ARC-Easy | ARC-Challenge | HellaSwag | PIQA | WinoGrande |
|---|---|---|---|---|---|---|
| AdamW DDP | 2.34 | 58.42% | 32.00% | 56.87% | 72.25% | 56.59% |
| DiLoCo | 2.37 | **60.48%** | 30.55% | 54.95% | **72.85%** | 55.56% |
| SparseLoCo | **2.36** | 59.05% | **32.17%** | **55.49%** | **72.85%** | **58.56%** |

## 4.3 ABLATIONS

We now highlight key design choices of SparseLoCo through a series of ablations.

**Outer Momentum + OuterEF**  A natural way to combine DiLoCo with OuterEF is by adding an error feedback while keeping the Nesterov outer optimizer. This has been attempted by Thérien et al. (2025), who showed it can provide benefits for quantization but that performance degrades quickly, despite using EF, with sparsification. This approach requires an additional accumulator. In contrast, our finding is that, in the case of high sparsification, using a global outer momentum can be detrimental to performance. This is illustrated in Table 6, where we equip SparseLoCo's outer optimizer with Nesterov outer momentum. This significantly degrades performance at high sparsity. We hypothesize that this is due to the conflicting directions of the error feedback and the outer momentum, since the largest components become amplified by the outer momentum but not the error feedback.

**Random-K**  We ablate the choice of TOP-$k$ compared to the alternative Random-$k$ (Shi et al., 2019; Wang et al., 2023) in Table 7. We observe that performance is significantly degraded when using random-$k$ for the same number of indices selected, emphasizing the importance of this design choice.

Table 6: Naively combining DiLoCo's standard Nesterov outer optimizer yields poor results. We use 3.12% communication density in both settings.

**Quantization**  As discussed, SparseLoCo benefits from stronger quantization than non-EF methods and supports up to 2-bit quantization and was generally observed to give results very close to full precision. In Table 7, we show the performance at different quantization values, showing that 2-bit quantization can be achieved at almost no performance cost.

| Method | Loss |
|---|---|
| SparseLoCo | 2.70 |
| SparseLoCo+Nesterov | 3.39 |

Table 7: **Ablation Studies** (*Left*): SparseLoCo with Random-$k$ vs. Top-k sparsification; Top-k significantly outperforms Random-$k$ across all communication densities. (*Right*): Effect of quantization on loss (lower is better); 2-bit shows almost no degradation vs. full precision.

| Density | Random-$k$ Loss | Top-k Loss |
|---------|-----------------|------------|
| 1.56% | 3.05 | 2.74 |
| 3.12% | 2.98 | 2.70 |
| 6.25% | 2.93 | 2.71 |

| Quant. bits | 1 | 2 | 3 | 4 | 32 |
|-------------|------|------|------|------|------|
| **Loss** | 4.79 | 2.70 | 2.70 | 2.70 | 2.70 |

## 5 CONCLUSION

We have proposed an algorithm that can blend multi-iteration LLM pre-training methods with Top-$k$ sparsification and quantization, enabling aggressive compression of DiLoCo's pseudo-gradients. Our work establishes that the outer momentum in DiLoCo can be replaced by local momentum accumulators without losing performance. Connecting local momentum with error feedback, we leverage this insight to develop *SparseLoCo*. Our extensive experiments confirm that SparseLoCo significantly reduces communication while outperforming strong baselines such as DiLoCo and DeMo, placing it on the Pareto frontier of loss versus communication volume. Additionally, our experiments reveal that sparse aggregation may actually be useful for improving the performances, opening the possibility of studying more sophisticated aggregation methods in the pre-training setting.

## 6 REPRODUCIBILITY STATEMENT

To facilitate reproducibility, we share the complete codebase with step-by-step instructions to replicate results for both our proposed method and the baselines, in the supplementary materials. Furthermore, in Appendix H and Tables 14 and 15, we report the hyperparameter sweep ranges and model architectural details in depth, with the selected configurations highlighted.

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

## A  REAL-WORLD DEPLOYMENT FOR COLLABORATIVE PERMISSIONLESS DISTRIBUTED TRAINING OVER THE INTERNET

SparseLoCo has been deployed in a real-world setting and is being used to collaboratively train models up to 8B and 70B with permissionless global participants using an incentive scheme Lidin et al. (2025) that rewards participants purely based on analysis of their compressed pseudo-gradients. This was done on top of an existing blockchain. The addition and coordination of peers and their rewards were handled through the blockchain. Communication of pseudo-gradients was routed through globally distributed, S3-compliant object storage—specifically Cloudflare R2—which enabled rapid dissemination of model updates worldwide. This setup allowed updates to be time-stamped and verified as part of the reward mechanism Lidin et al. (2025). Each peer maintained their own storage bucket, posting read credentials to a blockchain so that both other peers and the reward mechanism could access their compressed pseudo-gradients.

SparseLoCo is particularly advantageous in this communication setup, as cloud providers have large bandwidth for peer downloads and are able to rapidly distribute and mirror files across the globe. For upload, a peer only sends their pseudo-gradient through the cloud provider. Therefore, their outbound communication (and required upload bandwidth) is kept low. They then download the pseudo-gradients from the cloud provider, which is able to easily handle the high bandwidth constraints. An example of a practical communication time measured with an 8B model is on average 12 seconds, including sending their compressed pseudo-gradients and downloading other workers' messages with the test node never exceeding 500 Mb/s. Compared to processing with $8\times$ H200, which takes around 4.5 minutes, leading to minimal wall-clock time degradation despite traffic over the internet. For reference, Jaghouar et al. (2024), which trained a similarly sized model (10B) with 8-bit DiLoCo, reports a globally distributed all-reduce synchronization time of 8.3 minutes on average for a peak of $R=14$ nodes participating and processing time of 38 minutes. We also performed test measurements of communication time for a 70B model with the same setup as above ($R=20$ peers), measuring a total communication time of 70 seconds on average, with the test node never reaching more than 500 Mb/s downlink and 110 Mb/s uplink.

**70B LLM training** We have deployed SparseLoCo on the live system discussed above to train a 72B model, the largest collaborative foundational model training run ever considered. The deployment uses $R=20$ replicas each associated to a peer, $H=30$ inner steps and a global batch size of $\sim$8M tokens per inner step. Although the system design allows peers to use any target hardware that can achieve reasonable throughput, the suggested hardware requirements targeted an $8\times$ B200. Preliminary results on several benchmarks after an estimated 120B tokens have yielded bpb of 0.798 and Hellaswag (0-shot) downstream of 71.2% consistent with expectations at this scale, though no existing benchmarks at similar token budgets are available due to the scale of training.

## B  EFFECT OF CHUNKING, DCT, AND INNER STEPS

A recently introduced method Peng et al. (2024) considered the single-step setting with error feedback, using a compression function that first applies a discrete cosine transform (DCT) on tensor chunks and then selects the TOP-$k$ values in the DCT domain. It further employed sign descent on the final aggregated update Kunstner et al. (2023). Without the DCT transform, this approach can be seen as a special case of SparseLoCo when $H = 1$, the inner optimizer is plain SGD, and the outer optimizer utilizes sign descent. Since the effect of the DCT transform, designed for data with sequential structure where the order of elements matters, is not well understood in this context, and given the additional uncertainty about the role of chunking, in this section we disentangle the contributions of both for DeMo, as well as in the multi-step ($H > 1$) setting. The ablations of these three factors, evaluated purely in terms of loss, are presented in Table 8. Here, the TOP-$k$ EF baseline is a simplified DeMo that applies TOP-$k$ selection globally to the entire tensor (rather than within chunks) while still utilizing sign descent.

We observe that in the setting with no local steps (TOP-$k$ EF, DeMo) the impact of chunking is very significant and the performance of DeMo can be nearly recovered without resorting to the DCT. When using the local setting ($H>1$), we observed that DCT actually degrades performance; however, we also find that the impact of chunking is more limited than in the setting of $H=1$. We hypothesize that chunking and DCT both serve to reduce the effect of outlier values on the scale of

individual workers' contributions, which may be less critical in the case of SparseLoCo due to its adaptive inner optimizer.

Notably, DeMo does not have a natural way to incorporate adaptive optimization, and in practice, the sign descent is used to approximate the benefits of the Adam optimizer Peng et al. (2024). A significant advantage of SparseLoCo is that operating on the pseudo-gradients allows easy integration of adaptive optimizers like Adam in the inner loop.

Table 8: Ablation of tensor chunking and DCT (lower loss is better). We observe that chunking is critical for the performance of DeMo. With $H>1$, DCT degrades performance. All runs use full precision (FP32).

| Method | No DCT | DCT |
|---|---|---|
| SparseLoCo ($H > 1$, Chunking, TOP-$k$ EF) | **2.72** | 2.75 |
| SparseLoCo w/o Chunking ($H > 1$, TOP-$k$ EF) | 2.73 | 2.76 |
| DeMo (Chunking, TOP-$k$ EF w/ Sign Descent) | 2.87 | 2.83 |
| TOP-$k$ EF (w/ Sign Descent) | 3.48 | 2.84 |
| DiLoCo | 2.76 | – |

## C  OVERTRAINING REGIME WITH LARGE COMMUNICATION INTERVAL

Following Charles et al. (2025), we put SparseLoCo to the test in an overtraining regime by doubling the token budget to 20B and using a larger communication interval of $H=250$. Our observations are consistent with the trends in Figure 1, and SparseLoCo outperforms DILOCO at this setting (Table 9).

Table 9: Overtraining on $2\times$ data (20B token budget) with communication interval $H=250$.

| Method | Density | Loss |
|---|---|---|
| DiLoCo | 100% | 2.77 |
| SparseLoCo | 50% | **2.73** |
| SparseLoCo | 25% | 2.74 |
| SparseLoCo | 12.5% | 2.79 |
| SparseLoCo | 3.12% | 2.97 |
| SparseLoCo | 1.56% | 3.00 |

## D  STREAMING SPARSELOCO

In this section, we verify that Streaming DiLoCo Douillard et al. (2025) and SparseLoCo can be combined. Streaming DiLoCo is an orthogonal direction to SparseLoCo for reducing peak communication volume by hiding it. SparseLoCo reduces the absolute number of bits communicated per step through compression, thus indirectly reducing peak communication. Streaming DiLoCo directly reduces peak bandwidth by only communicating subsets of the model's parameters at a time but does not reduce the absolute number of bits communicated.

Table 10 reports results for combining SparseLoCo with Streaming DiLoCo to reduce peak communication volume. We train 18-layer 1B-parameter transformers (hidden dimension 2048) in this ablation. The model is partitioned into three even subsets of 6 hidden layers, with the first and third subsets containing the embedding and unembedding layers, respectively. We train the $1,055$M parameter model for a chinchilla-optimal 21B tokens Hoffmann et al. (2022). We use a communication interval of $H = 15$ for the full model (Streaming communicates every 5 steps) and 8 workers. We observe that both models reach the same final validation loss (it differed only in the 4th decimal), while Streaming SparseLoCo reduces peak communication volume by a factor of 3.

Table 10: We combine SparseLoCo with Streaming DiLoCo to reduce peak communication volume when training an 18-layer 1B parameter transformer. The model is partitioned into three even subsets of 6 hidden layers, with the first and third subsets containing the embedding and unembedding layers, respectively. We use a communication interval of $H = 15$ for the full model (Streaming communicates every 5 steps) and 8 workers. We observe that both models reach the same final validation loss (it differed only in the 4th decimal), while Streaming SparseLoCo reduces peak communication volume by a factor of 3.

| Method | Density | Comm. Volume/Step | Peak Comm. Volume | Loss |
|---|---|---|---|---|
| SparseLoCo | 3.125% | 35.03 MB | 35.03 MB | **2.51** |
| Streaming SparseLoCo | 3.125% | 35.03 MB | 11.68 MB | **2.51** |

# E    FREEZING ERROR FEEDBACK

We apply a short error feedback (OuterEF) freeze at the beginning of training: for the first few outer steps, the error feedback $e_r$ is not utilized. Concretely, during the freeze we don't use nor accumulate in the EF buffer. We find that freezing the OuterEF for the first few outer steps slightly improves training stability and overall performance (see Table 11).

Table 11: Freezing error feedback for the first few outer steps improves training. The final validation loss for 512M models trained with SparseLoCo (3.12% density), $R=8$ replicas, and communication interval $H=15$ is reported.

| EF Freeze | Loss |
|---|---|
| 0% | 2.704 |
| 5% | 2.699 |

# F    SCALING REPLICAS ACROSS DIFFERENT DENSITIES AND COMMUNICATION INTERVALS

We compare DiLoCo and SparseLoCo while varying the number of workers $R \in \{8, 16, 32\}$, communication intervals $H \in \{15, 50, 100\}$ using model sizes 178M and 512M, and report the final validation loss in Tables 12, 13, and 4. We observe that SparseLoCo outperforms DiLoCo with higher number of parallel workers.

Table 12: Final validation loss for the 178M model while varying the number of workers ($R \in \{8, 16, 32\}$) and the communication interval ($H \in \{15, 50, 100\}$). **Best** is bold.

**H=15**

| Method | Density | Loss R=8 | Loss R=32 |
|---|---|---|---|
| AdamW | 100.00% | 2.91 | 2.91 |
| DiLoCo | 100.00% | 2.99 | 3.10 |
| SparseLoCo | 0.78% | 2.96 | 3.02 |
| | 1.56% | 2.93 | 3.00 |
| | 3.12% | **2.91** | **2.99** |
| | 6.25% | 2.94 | 3.00 |
| | 12.50% | 2.96 | 3.04 |
| | 25.00% | 2.96 | 3.14 |
| | 50.00% | 3.03 | 3.29 |

**H=50**

| Method | Density | Loss R=8 | Loss R=32 |
|---|---|---|---|
| AdamW | 100.00% | 2.91 | 2.91 |
| DiLoCo | 100.00% | 3.05 | 3.20 |
| SparseLoCo | 0.78% | 3.09 | 3.13 |
| | 1.56% | 3.03 | **3.07** |
| | 3.12% | **2.99** | 3.09 |
| | 6.25% | **2.98** | 3.09 |
| | 12.50% | 3.00 | 3.14 |
| | 25.00% | 3.04 | 3.25 |
| | 50.00% | 3.12 | 3.42 |

**H=100**

| Method | Density | Loss R=8 | Loss R=32 |
|---|---|---|---|
| AdamW | 100.00% | 2.91 | 2.91 |
| DiLoCo | 100.00% | 3.12 | 3.29 |
| SparseLoCo | 0.78% | 3.20 | 3.29 |
| | 1.56% | 3.12 | 3.26 |
| | 3.12% | 3.05 | 3.19 |
| | 6.25% | **3.03** | **3.17** |
| | 12.50% | **3.03** | 3.21 |
| | 25.00% | 3.07 | 3.32 |
| | 50.00% | 3.17 | 3.48 |

Table 13: Final evaluation loss of scaling number of workers $R \in \{8, 16, 32\}$ for different communication interval $H \in \{15, 100\}$ using different communication densities for SparseLoCo using 512M model size. Best results in each communication interval are presented in **Bold**.

| Method | Density | Loss (R=8) | Loss (R=16) | Loss (R=32) |
|---|---|---|---|---|
| AdamW | 100.00% | 2.69 | 2.69 | 2.69 |
| | | | H=15 | |
| DiLoCo | 100.00% | 2.76 | 2.77 | 2.82 |
| | 0.78% | 2.79 | 2.81 | 2.84 |
| | 1.56% | 2.74 | 2.76 | 2.79 |
| SparseLoCo | 3.12% | **2.70** | **2.74** | **2.77** |
| | 6.25% | 2.71 | 2.76 | 2.78 |
| | 12.50% | 2.76 | 2.78 | 2.82 |
| | 25.00% | 2.77 | 2.78 | 2.93 |
| | | | H=100 | |
| DiLoCo | 100.00% | 2.87 | 2.94 | 3.05 |
| | 0.78% | 3.09 | 3.14 | 3.29 |
| | 1.56% | 3.02 | 3.06 | 3.21 |
| SparseLoCo | 3.12% | 2.96 | 3.03 | 3.12 |
| | 6.25% | 2.94 | 2.97 | 3.03 |
| | 12.50% | 2.85 | **2.88** | **3.02** |
| | 25.00% | **2.82** | 2.89 | 3.11 |

# G  COMPRESSION OF INDICES IN TOP-$k$

In TOP-$k$ methods, the indices of the selected values need to be transmitted alongside the values. When values are aggressively quantized (as in SparseLoCo), this index-transmission overhead becomes significant. In SparseLoCo, we utilize chunk sizes of $C$=4096, so, naively, we can transmit indices in 12 bits per transmitted value. However, with 2-bit quantization, this overhead becomes significant, motivating further index compression. Assuming a chunk size of $C$ and TOP-$k$ selection, we observe that the information-theoretic limit is $\log_2 \binom{C}{k}$ bits. For practical cases considered in this work ($C$=4096 and $k \in \{32, 128, 256\}$), this corresponds to 8.3, 6.3, and 5.3 bits per transmitted value, respectively. In practice, we designed a custom compression algorithm based on sub-chunking and coding that achieves 8.9, 6.6, and 5.6 bits per value for these cases.

# H  HYPERPARAMETER SELECTION DETAILS

Table 15 reports the hyperparameter search spaces for the 512M model size. We tune all methods at communication interval $H$=15 and reuse the best configurations for other settings; General and model architecture settings are fixed across all runs unless stated otherwise. In Table 14, we provide architectural details for 178M and 2B model sizes. For 178M, we reduce the batch size to 32 leading to an effective batch size of $524, 288$, and repeating the hyper-parameter sweeps as Table 15, we observe the same optimal settings. For the 2B model size, we increase warmup to $800$ and perform a small sweep of learning-rates lower than the optimal setting of 178M and 500M models. Specifically, for DiLoCo we search $\alpha_{\text{inner}} \in \{8\text{e}-4, 6\text{e}-4\}$ and $\alpha_{\text{outer}} \in \{0.6, 0.4\}$, finding $\alpha_{\text{inner}}$=8e−4, $\alpha_{\text{outer}}$=0.6 optimal; for SparseLoCo we search $\alpha_{\text{inner}} \in \{1\text{e}-3, 8\text{e}-4\}$ and $\alpha_{\text{outer}} \in \{0.8, 0.6\}$, finding $\alpha_{\text{inner}}$=1e−3, $\alpha_{\text{outer}}$=0.8 optimal. For number of workers $R > 8$ experiments, we ensure the same effective batch size used for $R$=8 by scaling the batch size accordingly.

Table 14: Model settings for 178M (left) and 2B (right) model scales.

| Parameter | Value |
|---|---|
| Total Parameters | 177,622,016 |
| Number of Layers | 9 |
| Hidden Size | 1,024 |
| Intermediate Size | 2,688 |
| Attention Heads | 8 |
| Vocabulary Size | 32,000 |
| FFN Activation | SwiGLU |

| Parameter | Value |
|---|---|
| Total Parameters | 1,972,759,040 |
| Number of Layers | 24 |
| Hidden Size | 2,560 |
| Intermediate Size | 7,680 |
| Attention Heads | 20 |
| Key-Value Heads | 5 |
| Vocabulary Size | 32,000 |
| FFN Activation | SwiGLU |

Table 15: Hyperparameter search spaces for the 512M-parameter model scale. Bold entries indicate the best settings. Model and general settings (top) are fixed across all runs. We tune all methods at $H=15$ and reuse the best hyperparameters when varying $H$. With higher number of workers $R$, DiLoCo's optimal setting remained the same whereas SparseLoCo enjoys slightly lower outer learning rate. The effective batch size is given per inner step across all workers.

| General Settings | Value |
|---|---|
| Token Budget | 10.26B |
| Effective batch size | 4,194,304 |
| Sequence length | 2048 |
| Local batch size | 256 |
| Workers $R$ | 8 |
| Warmup steps | 500 |
| Inner gradient clipping | 1.0 |
| LR Decay | Cosine |
| Inner optimizer | AdamW |

| Parameter | Value |
|---|---|
| Total Parameters | 512,398,848 |
| Number of Layers | 12 |
| Hidden Size | 1536 |
| Intermediate Size | 5,440 |
| Attention Heads | 12 |
| Vocabulary Size | 32,000 |
| FFN Activation | SwiGLU |

| Setting | Hyperparameter | Search Space |
|---|---|---|
| AdamW Baseline | $\alpha$ | 4e-4, 6e-4, 8e-4, 1e-3 **2e-3**, **3e-3**, **4e-3**, 6e-3 |
| DiLoCo - Nesterov Outer | $H=15, R \in \{8, 16, 32\}$ | |
| | $\alpha_{\text{inner}}$ | 4e-4, 6e-4, **8e-4**, 1e-3 |
| | $\alpha_{\text{outer}}$ | 0.2, 0.4, **0.6**, 0.8, 1.0 |
| | momentum | **0.9** |
| SparseLoCo (Density=0.78%) | $H=15, R=8$ | |
| | $\alpha_{\text{inner}}$ | 6e-4, 8e-4, **1e-3**, **2e-3**, 3e-3 |
| | $\alpha_{\text{outer}}$ | 0.4, 0.6, 0.8, **1.0** |
| | error momentum ($\beta$) | 0.9, **0.95**, 0.999 |
| | $H=15, R=16$ | |
| | $\alpha_{\text{outer}}$ | 0.6, **0.8**, 1.0 |
| | $H=15, R=32$ | |
| | $\alpha_{\text{outer}}$ | 0.4, **0.6**, 0.8 |
| DiLoCo - SGD Outer | $\alpha_{\text{inner}}$ | 6e-4, **1e-3** |
| | $\alpha_{\text{outer}}$ | 0.8, **1.0** |
| | momentum | **0.0** |
| DiLoCo-LOM | $\alpha_{\text{inner}}$ | 6e-4, **8e-4**, 1e-3 |
| | $\alpha_{\text{outer}}$ | 0.4, **0.6**, 0.8, 1.0 |
| | momentum | **0.9** |
| DeMo | $\alpha$ | 8e-4, **1e-3**, 3e-3 |
| | error momentum ($\beta$) | 0.95, **0.999** |

## I  CONVERGENCE

We now show a convergence guarantee for SPARSELOCO with inner Local Adam and error feedback (EF). Our argument builds directly on the high–probability analysis of Local Adam in Cheng & Glasgow (2025) and uses standard EF techniques for contractive compressors Karimireddy et al. (2019).

Recall the Local Adam setup of Cheng & Glasgow (2025): there are $M$ workers, $R$ communication rounds, and $K$ local steps per round, so each worker computes $T := KR$ stochastic gradients. We adopt their notation $(z_{r,k}, H_r)$ for the synchronized iterates and diagonal preconditioners, and write $\nabla f(z_{r,k})$ for the population gradient.

**SparseLoCo + Local Adam + EF.**  At the end of round $r$, worker $m$ has a Local–Adam direction

$$\Delta_m^{(r)} := H_{m,r}^{-1} u_{m,r}, \qquad H_{m,r} = \mathrm{diag}\big(\sqrt{v_{m,r} + \lambda^2}\big),$$

with $u_{m,r}, v_{m,r}$ the first/second moment accumulators as in Algorithm 1 of Cheng & Glasgow (2025). In SPARSELOCO, workers apply classical EF ($\beta = 1$) before communication:

$$\hat{\Delta}_m^{(r)} = \mathcal{C}\big(e_m^{(r)} + \Delta_m^{(r)}\big), \qquad e_m^{(r+1)} = e_m^{(r)} + \Delta_m^{(r)} - \hat{\Delta}_m^{(r)},$$

with $e_m^{(0)} = 0$. The all–reduce computes

$$\bar{s}^{(r)} := \frac{1}{M} \sum_{m=1}^{M} \hat{\Delta}_m^{(r)}, \qquad \bar{\Delta}^{(r)} := \frac{1}{M} \sum_{m=1}^{M} \Delta_m^{(r)}, \qquad \bar{e}^{(r)} := \frac{1}{M} \sum_{m=1}^{M} e_m^{(r)},$$

and the global iterate is updated as in Alg. 1:

$$z_{r+1,0} = z_{r,0} - \eta \, \bar{s}^{(r)},$$

with the same outer stepsize $\eta$ and all other hyperparameters as in (Cheng & Glasgow, 2025, Thm. 3 / Thm. C.3).

**Lemma 1** (EF telescoping identity). *For every round $r$,*

$$\bar{s}^{(r)} = \bar{\Delta}^{(r)} + \xi^{(r)}, \qquad \xi^{(r)} := \bar{e}^{(r)} - \bar{e}^{(r+1)}.$$

**Lemma 2** (EF residual bound for contractive compressors). *Assume the compressor $\mathcal{C}$ is $\omega$-contractive in mean square:*

$$\mathbb{E}\big\|\mathcal{C}(v) - v\big\|^2 \leq (1 - \omega) \|v\|^2, \qquad \omega \in (0, 1].$$

*(For deterministic TOP-$k$, $\omega = \frac{k}{d}$.) Then there exist absolute constants $C_e, C_\xi > 0$ such that, for all horizons $R \geq 1$,*

$$\sum_{r=0}^{R-1} \mathbb{E}\big\|\bar{e}^{(r)}\big\|^2 \leq C_e \frac{1 - \omega}{\omega^2} \sum_{r=0}^{R-1} \frac{1}{M} \sum_{m=1}^{M} \mathbb{E}\big\|\Delta_m^{(r)}\big\|^2,$$

$$\sum_{r=0}^{R-1} \mathbb{E}\big\|\xi^{(r)}\big\|^2 \leq C_\xi \frac{1 - \omega}{\omega^2} \sum_{r=0}^{R-1} \frac{1}{M} \sum_{m=1}^{M} \mathbb{E}\big\|\Delta_m^{(r)}\big\|^2, \qquad \xi^{(r)} := \bar{e}^{(r)} - \bar{e}^{(r+1)}.$$

Lemma 2 is the standard EF estimate for contractive compressors: the EF residuals are controlled, up to a factor $(1 - \omega)/\omega^2$, by the same quadratic budget $\sum_r \frac{1}{M} \sum_m \|\Delta_m^{(r)}\|^2$ that already appears in the Local Adam analysis of Cheng & Glasgow (2025).

We now recall Cheng–Glasgow's Local Adam bound in a compact notation. Let

$$\mathcal{G}_{\mathrm{LA}} := \frac{1}{\lambda} \tilde{\mathcal{O}}\Bigg( \frac{\tau \Delta}{R} + \frac{L\Delta}{KR} + \sqrt{\frac{L\Delta\sigma^2}{MKR}} + \frac{(L\Delta\sigma)^{2/3}}{K^{1/3}R^{2/3}} + \Bigg( \frac{L\Delta \, \sigma^{\alpha/(\alpha-1)}}{KR} \Bigg)^{\frac{2(\alpha-1)}{3\alpha-2}} \Bigg), \quad (1)$$

where the $\tilde{\mathcal{O}}(\cdot)$ hides the same logarithmic factors and dimension dependence as on the right-hand side of (Cheng & Glasgow, 2025, Eq. (4.9), Thm. 3 / Thm. D.3).

**Theorem 1** (SparseLoCo + Local Adam + EF preserves Cheng–Glasgow's rate). *Adopt the assumptions and hyperparameter conditions of **Theorem 3** (full **Theorem D.3**) in Cheng & Glasgow (2025) and let the compressor $\mathcal{C}$ be $\omega$-contractive. Run* SPARSELOCO *with inner Local Adam and EF as above, using the same outer stepsize $\eta$. Then there exists an absolute constant $C_{\mathrm{EF}} > 0$ (independent of $M, R, K$ and of all problem parameters) such that*

$$\frac{1}{KR} \sum_{r=0}^{R-1} \sum_{k=0}^{K-1} \mathbb{E}\big\|\nabla f(z_{r,k})\big\|_{H_r^{-1}}^2 \;\leq\; \left(1 + C_{\mathrm{EF}} \frac{1-\omega}{\omega^2}\right) \mathcal{G}_{\mathrm{LA}}. \tag{2}$$

*In particular, the dependence on $(M, R, K)$ and on all problem parameters is the same as in the Local Adam bound of **Theorem 3**Cheng & Glasgow (2025), up to the multiplicative factor $1 + C_{\mathrm{EF}} \frac{1-\omega}{\omega^2}$. Whenever $\mathcal{G}_{\mathrm{LA}} \to 0$ as $R, K$ grow (e.g., in the weakly convex regime of Cheng & Glasgow (2025)), the EF-augmented* SPARSELOCO *iterates satisfy the same vanishing-gradient guarantee.*

*Proof sketch.* The Local Adam proof of Cheng & Glasgow (2025) establishes a one-step descent inequality for the generalized Moreau envelope $f_{H_r}^\gamma$:

$$f_{H_{r+1}}^\gamma(z_{r+1,0}) \;\leq\; f_{H_r}^\gamma(z_{r,0}) \;-\; c_0 \eta \big\|\nabla f_{H_r}^\gamma(z_{r,0})\big\|_{H_r^{-1}}^2 \;+\; U_r,$$

where $c_0 > 0$ and $U_r$ collects the stochastic, clipping, and local–discrepancy terms. In their analysis, the update direction is the uncompressed average $\bar{\Delta}^{(r)}$. In SPARSELOCO, the update uses the EF direction $\bar{s}^{(r)} = \bar{\Delta}^{(r)} + \xi^{(r)}$ from Lemma 1. Substituting $\bar{s}^{(r)}$ into the same descent step only changes the alignment term and the quadratic smoothness term. Using Cauchy–Young inequalities together with the bounds on $H_r$ from (Cheng & Glasgow, 2025, Lem. D.4, Eq. (D.32)), we obtain an EF-modified one-step inequality of the form

$$\begin{aligned} f_{H_{r+1}}^\gamma(z_{r+1,0}) &\leq f_{H_r}^\gamma(z_{r,0}) - c_1 \eta \big\|\nabla f_{H_r}^\gamma(z_{r,0})\big\|_{H_r^{-1}}^2 \;+\; U_r \\ &\quad + C_1 \eta \big\|\xi^{(r)}\big\|^2 + C_2 \eta^2 \Big(\big\|\bar{\Delta}^{(r)}\big\|^2 + \big\|\xi^{(r)}\big\|^2\Big), \end{aligned}$$

for some absolute constants $c_1, C_1, C_2 > 0$.

Summing over $r = 0, \ldots, R-1$ on the same high–probability event as in (Cheng & Glasgow, 2025, Sec. 5, App. D) yields extra EF terms of the form

$$\sum_{r=0}^{R-1} \Big(\eta C_1 \|\xi^{(r)}\|^2 + \eta^2 C_2 \|\bar{\Delta}^{(r)}\|^2 + \eta^2 C_2 \|\xi^{(r)}\|^2\Big).$$

By Lemma 2, the EF residuals satisfy

$$\sum_{r=0}^{R-1} \mathbb{E}\big\|\xi^{(r)}\big\|^2 \;\leq\; C_\xi \frac{1-\omega}{\omega^2} \sum_{r=0}^{R-1} \mathbb{E}\big\|\bar{\Delta}^{(r)}\big\|^2,$$

so the total EF contribution is bounded by

$$C_{\mathrm{EF}} \frac{1-\omega}{\omega^2} \sum_{r=0}^{R-1} \eta^2 \,\mathbb{E}\big\|\bar{\Delta}^{(r)}\big\|^2,$$

for some absolute $C_{\mathrm{EF}} > 0$ (absorbing $C_1, C_2$ and the fixed stepsize $\eta$ used in Cheng & Glasgow (2025)).

The Local Adam proof already shows that the sum $\sum_r \eta^2 \mathbb{E}\|\bar{\Delta}^{(r)}\|^2$ is controlled by exactly the same quantity that yields the bound $\mathcal{G}_{\mathrm{LA}}$ (see the derivation of (Cheng & Glasgow, 2025, Eq. (4.9)) and its full version in their Theorem D.3). Thus, the EF contribution simply scales $\mathcal{G}_{\mathrm{LA}}$ by the factor $1 + C_{\mathrm{EF}} \frac{1-\omega}{\omega^2}$, leading to (2) after translating back from the envelope gradient to $\nabla f(z_{r,k})$ via (Cheng & Glasgow, 2025, Lem. D.4). $\square$

## J    Equivalence of LOM and Global Outer Momentum

We show that DiLoCo-LOM iterates are actually equivalent to DiLoCo.

**Proposition 1.** *Suppose identical initialization $m_r^{(0)} = m^{(0)} = 0$ for all $r \in [R]$, and fixed outer-momentum coefficient $\beta \in [0,1)$. Then, for all $t \geq 0$,*

$$\bar{m}^{(t)} = m^{(t)} \quad \text{and} \quad \bar{\tilde{\Delta}}^{(t)} = \tilde{\Delta}^{(t)},$$

*where $\bar{m}^{(t)} := \frac{1}{R} \sum_{r=1}^{R} m_r^{(t)}$ denotes the average of local momentum buffers, $\bar{\tilde{\Delta}}^{(t)} := \frac{1}{R} \sum_{r=1}^{R} \tilde{\Delta}_r^{(t)}$ the averaged LOM Nesterov direction, and $m^{(t)}$ and $\tilde{\Delta}^{(t)}$ the global momentum and Nesterov direction in DiLoCo, respectively. Consequently, the parameter updates of DiLoCo-LOM and DiLoCo are identical at every time step.*

*Proof.* We first show $\bar{m}^{(t)} = m^{(t)}$ by induction, then obtain $\bar{\tilde{\Delta}}^{(t)} = \tilde{\Delta}^{(t)}$ by linearity.

*Base case ($t = 0$).* With $m_r^{(0)} = 0$ and $m^{(0)} = 0$, we have $\bar{m}^{(0)} = \frac{1}{R} \sum_r m_r^{(0)} = 0 = m^{(0)}$.

*Inductive step.* Assume $\bar{m}^{(t-1)} = m^{(t-1)}$ for some $t \geq 1$. Averaging the local recursion,

$$\bar{m}^{(t)} = \frac{1}{R} \sum_{r=1}^{R} \left( \beta m_r^{(t-1)} + \Delta_r^{(t)} \right) = \beta \left( \frac{1}{R} \sum_r m_r^{(t-1)} \right) + \frac{1}{R} \sum_r \Delta_r^{(t)} = \beta \bar{m}^{(t-1)} + \bar{\Delta}^{(t)}.$$

By the global recursion, $m^{(t)} = \beta m^{(t-1)} + \bar{\Delta}^{(t)}$, hence $\bar{m}^{(t)} = m^{(t)}$. For the Nesterov directions,

$$\bar{\tilde{\Delta}}^{(t)} = \frac{1}{R} \sum_{r=1}^{R} \left( \Delta_r^{(t)} + \beta m_r^{(t)} \right) = \bar{\Delta}^{(t)} + \beta \left( \frac{1}{R} \sum_r m_r^{(t)} \right) = \bar{\Delta}^{(t)} + \beta \bar{m}^{(t)} = \bar{\Delta}^{(t)} + \beta m^{(t)} = \tilde{\Delta}^{(t)}.$$

## K    GPT-2 Experiments

To verify that SparseLoCo applies beyond LLaMA-style models, we also evaluate a 512M-parameter GPT-2 model. We reuse the best hyperparameters from the 512M LLaMA setting for both DiLoCo and SparseLoCo and train with $H = 15$ inner steps on the same dataset and token budget.

Table 16: **SparseLoCo vs. DiLoCo on a 512M-parameter GPT-2 model.** We report the final validation loss.

| Method | Density | Loss |
|---|---|---|
| SparseLoCo | 3.12% | 2.89 |
| DiLoCo | 100% | 2.92 |

## L    Compute Utilization vs. Bandwidth

We estimate the compute utilization of each method as $\frac{T_{\text{compute}}}{T_{\text{compute}} + T_{\text{comms}}}$, where $T_{\text{compute}}$ is the time spent in pure computation and $T_{\text{comms}}$ is the time spent in communication. We first estimate $T_{textcompute}$ considering the FLOPs profile of the model, assuming $8 \times$B200 GPUs per worker with $R = 16$ workers, a theoretical FP16/BF16 throughput of $4.5 \times 10^{15}$ FLOPs/s per GPU, and a reasonable machine FLOP utilization (MFU) of 40%. Then, we simulate training calculating $T_{\text{comms}}$ under different bandwidth constraints considering the pseudo-gradient message sizes of each method (Figure 3). At low bandwidths, SparseLoCo achieves substantially higher utilization than DDP, DeMo, and DiLoCo. For instance, at 1 Gbit/s, SparseLoCo exceeds 95% utilization, significantly outperforming the baselines.

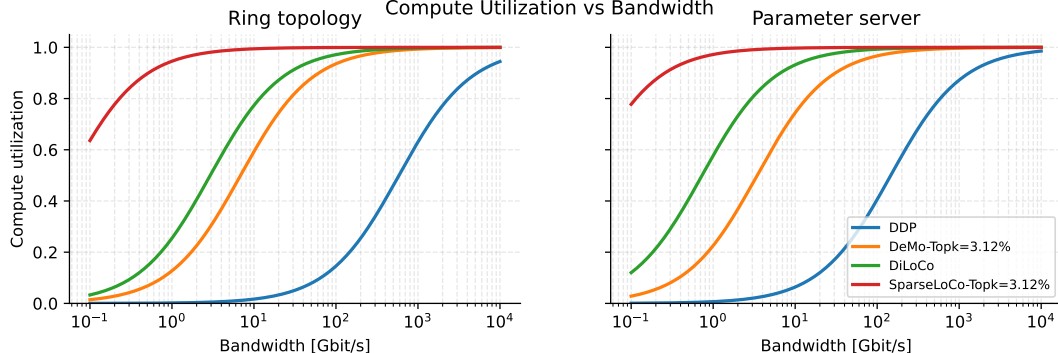

Figure 3: Compute utilization vs. bandwidth for DDP, DeMo, DiLoCo, and SparseLoCo under ring (left) and parameter-server (right) topologies. Compute utilization is calculated as total time spent in computation over full training time (including communication). We simulate a 70B LLaMA-2 model trained with $R=16$ replicas each with $8\times$ B200, assuming a reasonable 40% MFU under different bandwidth settings.

## M  SPARSITY ABLATIONS FOR DEMO AND SPARSELOCO

In Figure 4, we compare DeMo and SparseLoCo at different sparsity levels with DiLoCo, and DiLoCo without outer momentum. We train 512M models with $R=8$ replicas, and for the multi-step baselines, we use a fixed communication interval of $H=15$.

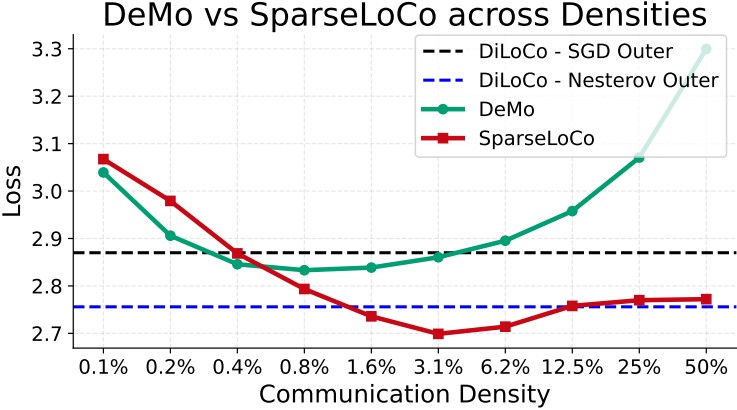

Figure 4: **DeMo and SparseLoCo across varying communication densities.** We compare DeMo and SparseLoCo across varying sparsity levels using the same settings as Figure 1; multi-step methods use a communication interval of $H=15$.

## N  LLM USAGE DISCLOSURE

We used large language models solely for language editing (grammar, vocabulary, and phrasing). All suggested edits were carefully reviewed by the authors before incorporating to the main text.

