# OpenReview forum: "Communication Efficient LLM Pre-Training with SparseLoCo"
_ICLR.cc/2026/Conference — Submitted to ICLR 2026_

### Official Review · Reviewer_Ag2i · 2025-10-27

**Soundness:** 2
**Presentation:** 3
**Contribution:** 2
**Rating:** 4
**Confidence:** 3

**Summary:**

This paper proposed a communication-efficient training algorithm, called SparseLoCo. By combining ideas of error feedback and compression, it can reach a low sparsity while outperforming DiLoCo baseline. Through comprehensive experiments, the authors further established the choice of sparsity hyperparameters, communication benefits, and scalability.

**Strengths:**

- The paper has clear motivation (Section 4.1) and comprehensive experiments, with a detailed hyperparameter search table.
- The proposed method, SparseLoCo outperforms DiLoCo when having a smaller pseudo-grad size.

**Weaknesses:**

- Both error feedback and gradient compression are already existing approaches, which raises concerns about novelty. It would be preferable to provide motivation and experimental justification for why this specific gradient compression scheme was selected.
- The paper fails to offer a reasonable explanation for why the method works. For example, on why SparseLoCo can outperform DiLoCo, the author refers to recent model-merging work in multi-task fine-tuning. But the setting in this work is pre-training, which should suffer from less accurate gradient information. And SparseLoCo introduces a new sparsity hyperparameter search space, which may limit its usage.

**Questions:**

1. Could you provide a possible explanation on why there's an optimal sparsity location in Figure 1? Is it related to the quantization scheme used? Is the optimal value is affected by the model size as well?
2. Comparing results from Table 3 and 5, does the improvement diminish as the model size grows larger?
3. Is it possible to provide ablation on the `Sparse` part of SparseLoco? How does the top-k contributes to the training overall?

---

> ### Author Response · Authors · 2025-11-21
>
> Thank you for your constructive feedback and comments. We are happy to hear you believe our paper is clearly motivated and that our experiments are comprehensive.
>
> ---
> ### W1: Novelty and choice of compression scheme
> Although error feedback and gradient compression are established in single-step settings, their combination in multi-step training (e.g., LocalSGD/DiLoCo) has been much less explored, especially in the context of LLM pre-training regimes where communication efficiency is of great interest in recent years [4.1-4.3]. A central goal of these approaches is to enable large-scale training in extremely bandwidth-constrained scenarios, such as over the internet or in cross-datacenter settings (see Introduction Paragraph 1, L32-39). However, reaching the most extreme compression ratios requires moving beyond purely quantization-based compression to sparsification, which previous work has been unable to get working in a DiLoCo context (see Figure 2 of [4.4] and Figure 11 of [4.3]).
>
> Our work’s biggest novelty lies in the non-trivial finding that replacing the DiLoCo’s outer momentum with a local error feedback accumulator allows us to aggressively sparsify the DiLoCo pseudogradient with no performance degradation (unlike [4.3-4.4]). We will now explain how we came to this finding. We would first like to emphasize that understanding how to apply the sparsification in the DiLoCo setting is not clear, and naively adding it on the outer gradients (as in [4.3-4.4]) while maintaining the Nesterov momentum leads to poor performance (see Table 6). As discussed in L471-473, this is due to tension between the error accumulator, which would reinforce the residual, while the Nesterov momentum would reinforce the top-k components (opposite of the residual). We find that a simple and non-trivial solution to this problem is not to use Nesterov momentum. Naively, one would expect degradation from not using outer momentum [4.5], but we show there is none and motivate it. Specifically in Sec 4.1, we present Diloco-LOM, a provably equivalent method to DiLoCo, which only uses a local outer momentum and reaches the same empirical performance as DiLoCo. Building on Diloco-LOM, we further demonstrate that maintaining the outer momentum on the bottom values (removing top-k) leads to no degradation compared to Nesterov momentum (Table 1). Thus leading us to the insight that error feedback, specifically with the top-k compressor, allows us to avoid Nesterov outer momentum altogether and produce a streamlined unified algorithm. Crucially, our algorithm comes at zero performance cost relative to standard DiLoCo while reaching unprecedented compression ratios.
>
> **Motivating the choice of compressor.** There are two popular compression methods considered in the literature: sparsification and quantization, with sparsification leading to the most aggressive compression since quantization is lower bounded by 1 bit per parameter. Comparing among sparsification approaches, we find that Top-k significantly outperforms random-k as per our ablations in Table 7.

---

> > ### Author Response · Authors · 2025-11-21
> >
> > ### W2: Why SparseLoCo works
> > We would like to first emphasize the clarification that W1 provides most of the motivation (which is supported in the paper, Sec. 4.1, as well as L194-198, and L351-356 of the original revision) of the method and why it is able to match Diloco without an outer momentum and with much less communication. Now, with regards to the improvement over Diloco, here we could only provide speculative suggestions, and we would like to clarify the connection to model merging methods. The similarity is that at the start of each communication round, all replicas share the same base model, and the resulting pseudo-gradients then take the same form as task vectors. Indeed, gradient information can be less accurate in pre-training, but recall the training synchronizes relatively frequently (e.g., every H=50 steps rather than after each replica has converged as in some model merging settings). Note as well that the referenced methods, such as TIES merging, are developed without any theory based on heuristic ideas like preventing interference through sparse aggregation, and this is what we are referencing here. Overall, we agree this is a heuristic suggestion (and indicate it as a hypothesis) and does not provide a full, theoretically satisfying explanation. On the other hand, we are confident our experiments demonstrate the improvement and include code and hyperparameter search grids. We also would like to emphasize that many of the design choices of Diloco are not well understood, for example, why Nesterov momentum vs other outer optimizers is so critical to its performance. Ultimately, we believe the importance of the Communication Efficient LLM pretraining problem to be high and thus, in turn, our contribution of proposing an algorithm that is at the Pareto frontier of existing methods on this problem is significant.
> >
> > Regarding the *extra sparsity* hyper-parameter, we would like to point out: Figure 2 demonstrates that most reasonable choices of the sparsity still lead to Pareto optimal solutions (better communications and better performance) than the existing SOTA baselines.
> >
> > ---
> > ### Question 1:
> > **(1) Why is there an optimal sparsity location in Fig. 1?**
> >
> >  As discussed in Sec. 4.2 (L359-365), three regimes emerge as density increases:
> > - At very high sparsity, almost no information is communicated, and performance degrades.
> > - At moderate densities, the aggregated pseudo‑gradients have sufficient information for a good update while the EF buffer remains non‑empty and *behaves like a sparsified outer momentum* with the highest components removed (we show in Sec 4.1 that this maintains the benefit of outer momentum); this is where SparseLoCo performs best.
> > - As density approaches 100%, line 12 in Alg. 1 nearly empties the EF buffer each step, effectively removing outer momentum; performance then approaches “DiLoCo without outer momentum,” which is clearly worse.
> >
> > This trade‑off naturally produces the “sweet spot” in Fig. 1. We have also added a new experiment plotting the validation loss against communication density for DeMo, observing the exact “optimal sparsity” behavior as SparseLoCo (Figure 4). We thank the reviewer for this question and have updated section 4.2 to provide a clearer explanation of the optimal sparsity phenomenon in Figure 1.
> >
> > **(2) Is optimal sparsity related to Quantization?** Our results show the optimal sparsity is **not** driven by the quantization scheme: Table 7 shows that 2‑bit quantization and full precision achieve almost identical loss, whereas 1‑bit clearly degrades performance.
> >
> >
> > **(3) Does optimal sparsity depend on model size?** Across 178M and 512M models (Tables 2, 5, 12, 13), the best densities consistently lie in a similar low‑percentage range. We do not observe a clear or systematic dependence on model size, while the dominant factor for optimal density is the communication interval H.

---

> > > ### Author Response · Authors · 2025-11-21
> > >
> > > ### Q2: Does the improvement diminish with larger models? (Tables 3 & 5)
> > > Considering the validation loss, SparseLoCo consistently matches or improves DiLoCo across scales.  We did not see a reduction in this gap going from 150M to 550M; thus, we would not make this conclusion. We also note that at 2B, due to the computational cost, we used a more restricted hyperparameter sweep. Nevertheless, we would like to re-emphasize that SparseLoCo is designed to reduce the communication volume. At 2B, even if we only match DiLoCo’s performance while communicating only 6.25% of the gradients is already a substantial practical improvement.
> > >
> > > ---
> > > ### Q3: Ablation on the “Sparse” part (role of Top‑k)
> > >
> > > We believe the reviewer is asking us to ablate the SparseLoCo with just sparsity and not error feedback. Here are the results in two settings:
> > > | Method                      | Loss |
> > > |-----------------------------|------|
> > > | SparseLoCo w/o EF (topk=3%) | 3.11 |
> > > | SparseLoCo (topk=3%)        | 2.91 |
> > >
> > > We also note that we have ablated random-k and various sparsity levels in the paper (Table 7 and Figure 1, respectively). Please let us know if there is another comparison that is requested.
> > >
> > >
> > > ---
> > > ### References
> > >
> > > [4.1] Charles, Zachary, et al. "Communication-Efficient Language Model Training Scales Reliably and Robustly: Scaling Laws for DiLoCo." arXiv preprint arXiv:2503.09799 (2025).
> > >
> > > [4.2] Ahn, Kwangjun, et al. "Dion: Distributed orthonormalized updates." arXiv preprint arXiv:2504.05295 (2025).
> > >
> > > [4.3] Douillard, Arthur, et al. "Streaming diloco with overlapping communication: Towards a distributed free lunch." arXiv preprint arXiv:2501.18512 (2025)
> > >
> > > [4.4] Thérien, Benjamin, et al. "MuLoCo: Muon is a practical inner optimizer for DiLoCo." arXiv preprint arXiv:2505.23725 (2025).
> > >
> > > [4.5] Douillard, Arthur, et al. "Diloco: Distributed low-communication training of language models." arXiv preprint arXiv:2311.08105 (2023).

---

> > > > ### Author Response · Authors · 2025-11-27
> > > >
> > > > Dear Reviewer Ag2i,
> > > >
> > > > Thank you for your valuable and constructive feedback. We have addressed all your questions and concerns in the revised paper and our detailed reply above. We made a substantial effort to incorporate your feedback, which we believe greatly strengthened the draft.
> > > >
> > > > If our reply has addressed your outstanding concerns, we ask you to please consider raising your score.
> > > >
> > > > Best regards,
> > > >
> > > > The Authors

---

### Official Review · Reviewer_aWwE · 2025-10-30

**Soundness:** 2
**Presentation:** 2
**Contribution:** 2
**Rating:** 2
**Confidence:** 3

**Summary:**

The authors propose SparseLoCo, which combines error feedback with TOP-k sparsification and 2-bit quantization to reduce communication volume to 1–3% of the original gradient, while maintaining performance superior to full-precision DiLoCo.

**Strengths:**

The authors achieve an exceptionally high gradient compression ratio, which reduces communication overhead, and conduct a detailed comparison with DiLoCo, demonstrating that their method maintains performance even under high sparsity.

**Weaknesses:**

- While the paper presents interesting ideas, its novelty compared to DeMO could be more clearly highlighted. It would be helpful if the authors more explicitly elaborated on the specific advancements beyond DeMO.
- The current writing could be further polished to improve overall readability and flow.

**Questions:**

- In subsection 3.3, the authors describe how SparseLoCo divides tensors into chunks and applies Top-k to each. Could you kindly provide more details on how this process is integrated into the framework?
- The terms "Density" and "Communication Density" are used without clear definitions.
- It would be valuable to know more about the concrete impact on training time—are there any quantitative results available?
- There seems to be a minor inconsistency: in the sentence "Concretely, the aggregation step in Algorithm 1 Line 14," "Line 14" may refer to "line 13" instead.
- Adding a more detailed comparison with DeMO, such as including its results in Figure 1, would strengthen the experimental evaluation.
- Could the authors provide a more comprehensive explanation of the theoretical convergence guarantees?

---

> ### Author Response · Authors · 2025-11-21
>
> We thank the reviewer for their feedback.
>
> ---
> ### W1: Novelty vs DeMo
> SparseLoCo is best viewed as a complement to DiLoCo: DiLoCo reduces the **frequency** of communication via local steps (H>1) but still sends full pseudo‑gradients, whereas SparseLoCo keeps this multi‑iteration DiLoCo/FedOpt setup, **aggressively reduces the volume** of each message, and crucially yields improved performance over a DiLoCo baseline. We emphasize that finding the way to integrate TopkEF with DiLoCo is not trivial, as shown by extensive ablations, and our approach yields a significant new algorithm. An a priori expectation would be that aggressive compression comes at a tradeoff, while our work *gives the surprising and significantly novel observation (not covered in any other work, either practical or theoretical) that in fact it does not and even improves results in many cases*.
>
> Our manuscript discusses DeMo and its relationship in multiple places (e.g., L118-L120), stating “DeMo (Peng et al., 2024) considers EF with DCT encoding and TOP-k compression in the LLM setting, demonstrating it can achieve competitive performance, but without incorporating local updates or the ability to leverage adaptive optimizers”. We note that SparseLoCo *does not use the DCT compressor* and leverages traditional Topk-EF. We also provide a detailed analysis of DeMo’s main technical contribution of the DCT compressor in Appendix B, which demonstrates its limitations, particularly, and includes direct comparisons to ablations of DeMo not provided in the original paper. Furthermore, we would like to emphasize the key empirical results in Table 2, which incorporate DeMo, clearly demonstrating the novelty: based on our tuned results (HP and code provided with submission), DeMo significantly underperforms both AdamW DDP and DiLoCo when all the baselines are well tuned. SparseLoCo outperforms both (and in particular DeMo) by a wide margin while being drastically more communication-efficient. Thus, we believe there is a clear novelty.  We would also like to mention that in the production environment and real world deployment discussed in Appendix A we did also evaluate DeMo and found that its communication overhead were more than 50% (with degraded performance) as it requires frequent synchronization, confirming the results of the paper illustrated in Figure 2 and Table 2, while SparseLoCo has communication overhead of less than 5%. We illustrate that this communication overhead reduction theoretically allows for a significantly higher compute utilization for SparseLoCo simulating training of a 70B model with different bandwidth constraints in the newly added Appendix L. We have now added the DeMo baseline to Figure 1 and put this in Appendix M, as requested. Overall, we believe that our paper performs an extensive comparison to both DeMo and DiLoCo.
>
> ---
> ### W2: Writing clarity
> We thank the reviewer for this comment. We appreciated the concrete suggestions provided in the questions and have revised the paper accordingly, addressing each question and point of clarification. We hope these changes further improve the overall readability and flow. If there are any additional parts the reviewer feels would benefit from further clarification, we would be grateful for the feedback and will gladly refine the text.

---

> ### Author Response · Authors · 2025-11-21
>
> ### Q1: Chunk‑wise TOP‑k integration
> In SparseLoCo (Alg. 1, lines 10–13), we first update the local error‑feedback buffer for each tensor using the full dense pseudo‑gradient (line 10). For compression, each 2D parameter tensor (e.g., Transformer linear and attention weights) is partitioned into disjoint 64×64 blocks, and 1D tensors (e.g., layer norms) into contiguous chunks of size 4096; we then apply Top‑k independently within each chunk. This chunk-wise application of the compressor follows the standard practices in gradient compression and quantization [3.1-3.3]. The selected values and their (chunk‑local) indices are quantized, transmitted to the other workers (lines 11–13), and removed from the local EF buffer (line 12). We have updated Sec. 3.3 to clarify the chunk-wise Top-k integration.  Finally, we would like to refer the reviewer to the last paragraph of Section 3.3 (L205-212), where the benefits of this chunk-wise design are discussed.
>
> ---
> ### Q2: Meaning of “Density” and “Communication Density”
> They both refer to the same quantity, and we use density as a concise shorthand for “communication density”: the fraction of pseudo‑gradients transmitted at each outer step, i.e., the density of the sparse vector sent in lines 11–13 of Algorithm 1. Thus, the lower the density, the lower the communication volume. We have updated the paper to define these terms explicitly in Sec. 3.3 (L198-200) and in the caption of Table 2.
>
> ---
> ### Q3: Quantitative impact on training time
> In the newly added Appendix L, we provide a theoretical compute utilization against different bandwidth budgets for all methods. In this experiment, we simulate training of a 70B LLaMA-2 model, and calculate the percentage of training time spent in pure compute. In Figure 3, we observe that SparseLoCo significantly outperforms other methods in compute utilization, for instance, exceeding 97% compute utilization in the realistic over-the-internet 1Gbits/s bandwidth. We further demonstrate a strong practical evidence for the extremely low communication overhead of SparseLoCo in a real-world production deployment as follows:
> Appendix A reports wall‑clock measurements from a real‑world, over-the-internet deployment using SparseLoCo. For an 8B model, one outer step takes ≈12 s communication (upload + downloads of compressed pseudo‑gradients) versus ≈4.5 min compute, with the node’s bandwidth never exceeding 500 Mb/s; for a 70B model with R=20 peers, we measure ≈70 s communication per step under similar bandwidth limits. In contrast, Jaghouar et al. [3.4], with 8‑bit DiLoCo, report ≈8.3 min synchronization vs 38 min compute for a 10B model, so communication is a much larger fraction of wall‑clock time there.
>
> ---
> ### Q4: Line‑number inconsistency in Algorithm 1
> Thank you for bringing this to our attention. The sentence should refer to the aggregation step in Algorithm 1, line 13 (averaging the sparse updates), rather than line 14 (parameter update), and has been corrected in the revision (L410).
>
> ---
> ### Q5: DeMo in Figure 1
> We have added Appendix M, where we evaluate SparseLoCo and DeMo under various sparsity settings (Figure 4). We did not include DeMo in the original Figure 1, since the purpose of that figure is to illustrate the behavior of SparseLoCo with respect to H and its relationship to sparsity, whereas H is not defined for single-step methods such as DeMo. Indeed, we would like to highlight that Figure 2 is a visualization of all of these data points (including those from DeMo), showing both the performance and communications, thereby enabling a direct comparison.
>
> ---
> ### Q6: Theoretical guarantees
> We have now added a result in the Appendix I that extends recent results on LocaAdam to our setting with the outer error feedback.
>
>
>
> ---
> ### References
> [3.1] Dettmers, Tim, et al. "8-bit optimizers via block-wise quantization." arXiv preprint arXiv:2110.02861 (2021).
>
> [3.2] Li, Bingrui, Jianfei Chen, and Jun Zhu. "Memory efficient optimizers with 4-bit states." Advances in Neural Information Processing Systems 36 (2023): 15136-15171.
>
> [3.3] Peng, Bowen, Jeffrey Quesnelle, and Diederik P. Kingma. "Decoupled momentum optimization." arXiv preprint arXiv:2411.19870 (2024).
>
> [3.4] Jaghouar, Sami, et al. "Intellect-1 technical report." arXiv preprint arXiv:2412.01152 (2024).

---

> > ### Author Response · Authors · 2025-11-27
> >
> > Dear Reviewer aWwE,
> >
> > Thank you for your valuable and constructive feedback. We have addressed all your questions and clarified some misunderstandings in the revised paper and our detailed reply above. We made a substantial effort to incorporate your feedback, which we believe greatly strengthened the draft.
> >
> > If our reply has addressed your outstanding concerns, we ask you to please consider raising your score.
> >
> > Best regards,
> >
> > The Authors

---

### Official Review · Reviewer_sx44 · 2025-10-31

**Soundness:** 2
**Presentation:** 3
**Contribution:** 2
**Rating:** 4
**Confidence:** 2

**Summary:**

This paper investigates communication-efficient training algorithms and proposes a method named SparseLoCo. The algorithm incorporates the error-feedback technique, specifically by communicating the top-k error updates across machines. Experiments conducted on the LLaMA model demonstrate the efficiency of the proposed approach.

**Strengths:**

The paper is well-written and the method is clearly presented. The experiments convincingly validate that the proposed approach effectively reduces communication overhead.

**Weaknesses:**

I have major concerns regarding the novelty and experimental validation of this work, which I find to be incremental.

1. Limited Novelty: The core technique of error-feedback is a well-established standard for achieving communication efficiency. The paper does not convincingly demonstrate a significant algorithmic advancement beyond this.

2. Insufficient Evidence for Acceleration: The experiments fail to prove that the method accelerates standard LLM training in practical settings (e.g., on 8 or 32 GPU platforms). Crucially, there is no plot showing the training loss against wall-clock time, which is essential for validating actual speedup.

3. Lack of LLM-Specific Innovation: While focused on LLM training, the method appears to be a generic application of sparsification and error-feedback, with no special design or adaptation for Transformer architectures.

Given the incremental nature of the contribution, I would expect a more compelling validation, such as a high-quality code package and solid experiments demonstrating clear wall-clock time acceleration in standard LLM training benchmarks.

**Questions:**

1. How does the proposed method compare to the standard baseline in terms of competitive wall-clock time for model convergence on a common scale, such as an 8-GPU platform?

2. Does the algorithm incorporate any domain-specific optimizations tailored for the Transformer architecture, or is it a generic approach?

3. To better demonstrate generality, could the authors show the method's performance on other model families, such as ResNet or GPT-2?

---

> ### Author Response · Authors · 2025-11-21
>
> We would like to thank the reviewer for their detailed comments. We are pleased that the reviewer finds our paper well-written, clearly presented, and that our experiments convincingly validate that our method reduces communication overhead.
>
> ---
> ### W1: Limited novelty
> We respectfully disagree. Our paper makes substantial novel contributions. Specifically, we introduce the first practical algorithm combining top-k error feedback with multi-iteration distributed optimization (building on methods like DiLoCo) for a foundational model training setting. This combination achieves the surprising result of simultaneously reducing communication and improving training performance—a dual improvement unprecedented in the literature and with significant practical implications. Through significant-scale experiments, we demonstrate SparseLoCo’s ability to permit LLM training in low-bandwidth environments.
>
> ---
> ### W2 & Q1: Evidence for acceleration/wall-clock time
> We would like to first reiterate the setting that our paper addresses. As discussed in the first paragraph of the introduction, we focus on the highly relevant bandwidth-constrained setting of cross-data-center training and training over the internet. This setting has received significant interest in recent years, with a number of papers accepted at major ML conferences this year [2.1-2.4]. Our setting is not designed for 8 GPU nodes with high-bandwidth interconnects.
>
> Taking this into account and pointing the reviewer to our Appendix as well, we would like to respectfully disagree: we believe the evidence for practical benefits we show is strong and does clearly demonstrate acceleration in practical settings:
>
> 1. We highlight that the benefits of any communication-efficient method are related to the computing environment, which includes both the bandwidth between nodes and the underlying communication primitives. Our method can operate in cross-datacenter settings [2.5] and over the internet settings [2.6]. In each of these cases, it would give a benefit as shown in Figure 2: SparseLoCo can support much lower bandwidths while achieving higher performance! This directly implies wall clock time savings. To highlight this, we now further expand this analysis in Appendix L, showing the improvements in theoretical compute utilization under different cross-node bandwidths (Figure 3). For instance, under realistic over-the-internet settings, 1Gbps, SparseLoCo significantly outperforms other baselines by a wider margin.
> 2. We would like to point the reviewer to Appendix A of the paper, which discusses acceleration and bandwidth constraints and wall clock times in a real-world *production deployment setting* and from *the largest publicly reported pre-training run ever conducted over the internet*, using SparseLoCo. This section discusses both a direct quantitative comparison showing better compute utilization than Jaghouar et al’s production run and the practical deployment of a model up to 72B, which would be impossible (more communication than compute) without a communication-efficient approach.
> 3. We also emphasize that this analysis approach is more comprehensive than picking an arbitrary compute environment and is standard in the relevant literature.

---

> > ### Author Response · Authors · 2025-11-21
> >
> > ### W3 & Q2: The technique is a generic application of Sparsification
> > We respectfully disagree with the reviewer. **It is demonstrably false to state that the technique is a generic application of sparsification to DiLoCo.** Previous works [2.1] and [2.7] have directly or *generically* applied sparsification to compress DiLoCo’s pseudogradient, resulting in a significant degradation in performance at low sparsities (see Figure 11 of [2.1] and Figure 2 of [2.7]). In contrast, we develop a subtle methodology explained in sections 3.1, 3.2, and 3.3 that allows us to aggressively sparsify the DiLoCo pseudogradient with no performance degradation (unlike [2.1,2.7]). Relative to existing work on DiLoCo [2.1-2.2,2.7-2.10], our key insight is that outer momentum can be locally approximated and that Nesterov momentum conflicts with error feedback. These previously unknown findings allow us to move beyond the difficulties encountered in [2.1] and [2.7], making a significant contribution to the field.
> >
> > **Lack of LLM-Specific Innovation/no special design or adaptation for Transformer architectures** We respectfully disagree with the reviewer that this is a weakness. As highlighted in the paragraph above, we make a clear and non-trivial contribution to the field of communication-efficient LLM pre-training. All our experiments were performed on LLM tasks, and as such, our technique is directly applicable to LLMs. In fact, SparseLoCo defines the Pareto frontier between communication volume and performance in LLM pre-training (see Figure 2).
> >
> > Although SparseLoCo can be used in other problems, we have developed and refined it in the critical context of LLM pre-training. It is well established in the communication efficiency literature that communication-efficient methods can perform differently on different problem classes. A relevant example of this is Local SGD, which did not work on CNNs and ImageNet  [2.11] but was adapted to DiLoCo for LLMs, showing great practical utility.  Indeed, just this year, many publications at top-tier venues such as NeurIPS show communication-efficient optimization methods developed specifically in the context of LLMs (but which do not have a clear transformer-specific element as the reviewer suggests): for example [2.1,2.2,2.10], demonstrating the clear relevance of this problem setting to the community.
> >
> > ---
> > ### Q3: Other model families (ResNet, GPT‑2)
> > Because SparseLoCo operates purely at the pseudo‑gradient level, it can be applied to architectures such as ResNets and GPT‑2 without modification. We appreciate the reviewer’s suggestion and, to demonstrate generality beyond LLaMA‑style models, we have added a new experiment with a 512M‑parameter GPT‑2 model. We reuse the best settings of LLaMA-512M for both SparseLoCo and DiLoCo and evaluate these methods with H=15 inner steps for the same token budget on the same dataset:
> > - SparseLoCo (sparsity: 3.12%): 2.89
> > - DiLoCo: 2.92
> >
> > This new experiment is now added to the paper in the revision (Appendix K, Table 16). Since all our baselines are exclusively developed and evaluated in the LLM pre‑training setting, we restrict our experiments to this domain and leave CNN architectures such as ResNet for future work.

---

> > > ### Author Response · Authors · 2025-11-21
> > >
> > > ### Additional validation & code
> > > **High‑quality code package.** All experiments in the paper—including DiLoCo, DeMo, and DDP baselines—are already fully reproducible from the code and scripts provided in the supplementary material, with detailed hyperparameter ranges and model configurations reported in Appendix H. Upon acceptance, we will release this as a polished open‑source package.
> > >
> > > **Solid experiments.** The paper includes an extensive experimental study that is at larger scale than most of those discussed in related work: multiple model sizes (178M, 512M, 2B), scaling over the number of workers and communication intervals, detailed ablations (outer momentum interaction with EF, TOP‑k vs Random‑k, quantization levels, chunking/DCT, streamingDiLoCo), and now additional GPT‑2 experiments. These are significant experimental scales with only a handful of industrial labs that would have the resources to run even larger experiments. We also discuss a real‑world decentralized deployment with a 72B model (Appendix A), explicitly reporting communication times and showing that SparseLoCo dramatically reduces bandwidth requirements and communication wall‑clock relative to DiLoCo‑style baselines in realistic over-the-Internet settings. Taken together, we believe this already constitutes both a solid and comprehensive empirical validation.
> > >
> > > ---
> > > ### References
> > > [2.1] Douillard, Arthur, et al. "Streaming diloco with overlapping communication: Towards a distributed free lunch." arXiv preprint arXiv:2501.18512 (2025).
> > >
> > > [2.2] Charles, Zachary, et al. "Communication-Efficient Language Model Training Scales Reliably and Robustly: Scaling Laws for DiLoCo." arXiv preprint arXiv:2503.09799 (2025).
> > >
> > > [2.3] Ramasinghe, Sameera, et al. "Mixtures of Subspaces for Bandwidth Efficient Context Parallel Training." The Thirty-ninth Annual Conference on Neural Information Processing Systems.
> > >
> > > [2.4] Ramasinghe, Sameera, et al. "Protocol Models: Scaling Decentralized Training with Communication-Efficient Model Parallelism." arXiv preprint arXiv:2506.01260 (2025).
> > >
> > >
> > > [2.5] Douillard, Arthur, et al. "Diloco: Distributed low-communication training of language models." arXiv preprint arXiv:2311.08105 (2023).
> > >
> > >
> > > [2.6] Jaghouar, Sami, et al. "Intellect-1 technical report." arXiv preprint arXiv:2412.01152 (2024).
> > >
> > >
> > > [2.7] Thérien, Benjamin, et al. "MuLoCo: Muon is a practical inner optimizer for DiLoCo." arXiv preprint arXiv:2505.23725 (2025).
> > >
> > >
> > > [2.8] Liu, Bo, et al. "Asynchronous local-sgd training for language modeling." arXiv preprint arXiv:2401.09135 (2024).
> > >
> > > [2.9] Khaled, Ahmed, et al. "Understanding outer optimizers in local sgd: Learning rates, momentum, and acceleration." arXiv preprint arXiv:2509.10439 (2025).
> > >
> > > [2.10] Nabli, Adel, et al. "Acco: Accumulate while you communicate, hiding communications in distributed LLM training." (2024).
> > >
> > > [2.11] Ortiz, Jose Javier Gonzalez, et al. "Trade-offs of local sgd at scale: An empirical study." arXiv preprint arXiv:2110.08133 (2021).

---

> > > > ### Author Response · Authors · 2025-11-27
> > > >
> > > > Dear Reviewer sx44,
> > > >
> > > > Thank you for your valuable and constructive feedback. We have addressed all your questions and clarified some misunderstandings in the revised paper and our detailed reply above. We made a substantial effort to incorporate your feedback, which we believe greatly strengthened the draft.
> > > >
> > > > If our reply has addressed your outstanding concerns, we ask you to please consider raising your score.
> > > >
> > > > Best regards,
> > > >
> > > > The Authors

---

### Official Review · Reviewer_AinH · 2025-10-31

**Soundness:** 3
**Presentation:** 2
**Contribution:** 3
**Rating:** 4
**Confidence:** 3

**Summary:**

This paper introduces SparseLoCo, a communication-efficient algorithm designed for distributed LLM pre-training in bandwidth-constrained environments. The method addresses the limitations of some communication algorithms, which still suffer from transmitting large, full-precision pseudo-gradients. SparseLoCo innovatively combines DiLoCo's multi-step local updates with aggressive compression, integrating TOP-k sparsification (achieving densities as low as 1-3%) and 2-bit quantization. The core insight is that DiLoCo's global outer momentum can be effectively replaced by a local error feedback (OuterEF) accumulator, which naturally manages the local pseudo-gradient updates. This unification allows SparseLoCo to drastically reduce communication volume while simultaneously outperforming the full-precision DiLoCo baseline in terms of final model performance and communication-loss trade-offs.

**Strengths:**

1. The paper presents a novel and highly practical solution (SparseLoCo) that effectively unifies two distinct lines of communication-efficient training: infrequent communication (like DiLoCo) and aggressive gradient compression (sparsification + quantization). This is a significant contribution for training in highly bandwidth-constrained settings, such as cross-datacenter or internet-based collaboration.

2. The core insight that DiLoCo's global outer momentum can be successfully replaced by a local error feedback (OuterEF) mechanism is a strong and non-obvious contribution. This discovery is what enables the integration of aggressive sparsification, as the EF buffer naturally manages the compression error—a problem that previously hindered attempts to combine DiLoCo with sparsification.

3. The method is backed by compelling empirical evidence. SparseLoCo not only matches but outperforms the full-precision DiLoCo baseline in terms of final loss and achieves a superior position on the communication-loss Pareto frontier. The ability to do this with extreme compression levels (e.g., 1-3% density and 2-bit quantization) is impressive and demonstrates the practical viability of the approach.

**Weaknesses:**

1. The paper would benefit from a brief theoretical analysis clarifying the role of the local inner updates ($H$ steps) in the optimization process. The authors compare SparseLoCo to DeMo, which (despite also updating non-dominant information locally) provides a theoretical justification that convergence can be achieved even with minimal global information, as long as it represents the dominant components of the momentum. SparseLoCo explicitly designs a local inner loop before processing and communicating dominant components. Therefore, the impact of these inner loops on global convergence, and how they interact with the OuterEF and sparse aggregation, is a key aspect that deserves further theoretical exploration. Some brief analysis may help to understand the effect of $H$.

2. The experimental setting for Table 3 is not clearly described in the text, making it difficult to ascertain the model scale and training configuration used for these downstream benchmarks. Furthermore, to sufficiently demonstrate generalizability, the evaluation would be more convincing if it included a wider array of tasks. Given that the paper reports zero-shot accuracy, a discussion on the method's applicability and performance in the fine-tuning regime would also be a valuable and relevant addition, but this is not explored.

**Questions:**

See weaknesses.

---

> ### Author Response · Authors · 2025-11-21
>
> Thank you for the detailed and positive assessment of our contributions.
>
> ---
> ### W1: Inner updates, theory, and comparison to DeMo
> First, we clarify that the **DeMo** paper (which is a preprint on ArXiv [1.11]) does *not* provide a convergence result that the reviewer attributes to it. DeMo is an EF compressor that applies the DCT followed by the top-k operation in a single‑step regime (see L119-121). In contrast, SparseLoCo uses the classical Topk compressor with EF. The convergence of such EF+Top-k methods has been established in a number of works [1.1-1.4]. Similarly, the convergence of multi-step methods has been established [1.5-1.7]. We show in an updated appendix how the most recent results of Cheng & Glasgow [1.7] can also be extended and combined with recent results of LocalAdam to show a convergence result relevant for SparseLoCo.  We also refer the reviewer to Appendix B, where we explicitly analyze the DCT compressor in our setting and provide some novel insights on its importance.
>
> We emphasize that our work builds on top of the DiLoCo/FedOpt. DiLoCo already uses local inner steps (H>1); Furthermore, as discussed in the related work, there is a long literature on multi-iteration methods. We highlight, however, that despite being used in a number of algorithms in distributed and federated learning and other areas and despite many convergence results, theoretical analysis showing communication reduction in terms of H remains a challenging problem in this area, with only limited work recently making progress [1.7-1.8]. Those works focus on purely theoretical challenges, while ours brings a new method that is shown to work in a very relevant practical setting. Thus, we believe that a detailed theoretical analysis of H is well beyond the scope of this work. Nevertheless, as part of the rebuttal, we have extended the most recent result of Diloco style multi-iteration methods to the SparseLoCo setting, showing that state-of-the-art theoretical results with respect to H, among other things, are maintained by SparseLoCo (see Appendix I).
>
> ---
> ### W2: Table 3: setup and scope of evaluation
> **Table 3 setup.**
>
> We thank the reviewer for pointing out confusion related to Table 3. We note that we indicated on lines L211-215 (of the original revision) the setting for all the main results (as well as Appendix Chapter H and Tables 14 and 15). We completely agree that this can be clearer, and we have made additional clarifications in the caption of Tables 2 and 3.
>
> **Why not fine-tuning?**
>
> LLM pre-training is an extremely large-scale problem these days, and this has led to methods focused on communication efficiency of this setting to allow scaling distributed learning (for example, see [1.9-1.11]). We agree that finetuning is also an important setting, but in practice, it requires less distribution and is much less compute-intensive (thus, limited need for cross-datacenter or over-the-internet training). Moreover, it is known that completely different classes of methods can be effective for communication efficiency [1.12-1.15].

---

> > ### Author Response · Authors · 2025-11-21
> >
> > ---
> > ### References
> > [1.1] Karimireddy, Sai Praneeth, et al. "Error feedback fixes signsgd and other gradient compression schemes." International conference on machine learning. PMLR, 2019.
> >
> > [1.2] Stich, Sebastian U., and Sai Praneeth Karimireddy. "The error-feedback framework: Better rates for SGD with delayed gradients and compressed communication." arXiv preprint arXiv:1909.05350 (2019).
> >
> > [1.3] Richtárik, Peter, Igor Sokolov, and Ilyas Fatkhullin. "EF21: A new, simpler, theoretically better, and practically faster error feedback." Advances in Neural Information Processing Systems 34 (2021): 4384-4396.
> >
> > [1.4] Gruntkowska, Kaja, Alexander Tyurin, and Peter Richtárik. "EF21-P and friends: Improved theoretical communication complexity for distributed optimization with bidirectional compression." International conference on machine learning. PMLR, 2023.
> >
> > [1.5] Stich, Sebastian U. "Local SGD converges fast and communicates little." arXiv preprint arXiv:1805.09767 (2018).
> >
> > [1.6] Basu, Debraj, et al. "Qsparse-local-SGD: Distributed SGD with quantization, sparsification and local computations." Advances in Neural Information Processing Systems 32 (2019).
> >
> > [1.7] Cheng, Ziheng, and Margalit Glasgow. "Convergence of Distributed Adaptive Optimization with Local Updates." arXiv preprint arXiv:2409.13155 (2024).
> >
> > [1.8] Li, Tian, et al. "Federated optimization in heterogeneous networks." Proceedings of Machine learning and systems 2 (2020): 429-450.
> >
> > [1.9] Douillard, Arthur, et al. "Diloco: Distributed low-communication training of language models." arXiv preprint arXiv:2311.08105 (2023).
> >
> > [1.10] Douillard, Arthur, et al. "Streaming diloco with overlapping communication: Towards a distributed free lunch." arXiv preprint arXiv:2501.18512 (2025).
> >
> > [1.11] Peng, Bowen, Jeffrey Quesnelle, and Diederik P. Kingma. "Decoupled momentum optimization." arXiv preprint arXiv:2411.19870 (2024).
> >
> > [1.12] Malladi, Sadhika, et al. "Fine-tuning language models with just forward passes." Advances in Neural Information Processing Systems 36 (2023): 53038-53075.
> >
> > [1.13] Xu, Lingling, et al. "Parameter-efficient fine-tuning methods for pretrained language models: A critical review and assessment." arXiv preprint arXiv:2312.12148 (2023).
> >
> > [1.14] Liu, Han, et al. "EcoLoRA: Communication-Efficient Federated Fine-Tuning of Large Language Models." arXiv preprint arXiv:2506.02001 (2025).
> >
> > [1.15] Babakniya, Sara, et al. "Slora: Federated parameter efficient fine-tuning of language models." arXiv preprint arXiv:2308.06522 (2023).

---

> > > ### Author Response · Authors · 2025-11-27
> > >
> > > Dear Reviewer AinH,
> > >
> > > Thank you for your valuable and constructive feedback. We have addressed all your questions and clarified some misunderstandings in the revised paper and our detailed reply above. We made a substantial effort to incorporate your feedback, which we believe greatly strengthened the draft.
> > >
> > > If our reply has addressed your outstanding concerns, we ask you to please consider raising your score.
> > >
> > > Best regards,
> > >
> > > The Authors

---

### Author Response · Authors · 2025-12-03

Dear AC,

SparseLoCo represents a significant methodological contribution that leads to a state-of-the-art algorithm (Pareto optimal in communication and performance) for LLM pre-training under low bandwidth, across datacenters, and over the internet. Although our initial scores are relatively low, we believe the paper has not received a completely justified set of reviews and would request the AC to give it a full assessment.

**TLDR:** The reviewers misunderstand our experimental setting, resulting in concerns of limited novelty across the board. Other concerns include (A) missing theory regarding local steps, (B) insufficient explanations of efficiency, and (C) an insufficient explanation of performance improvement for DiLoCo. We provide thorough responses to each reviewer and update the manuscript, providing (A) new theory regarding convergence and local steps in section (*I*), (B) a section (*L*) describing SparseLoCo’s efficiency improvements in compute utilization under bandwidth constraints, and (C) an improved explanation of SparseLoCo’s improvement over DiLoCo (Lines 351-365).

**Reviewer’s confusion about limited novelty:** Although we use error feedback as part of the core method, we emphasize that the paper demonstrates a surprising and completely novel empirical property in the setting of DiLoCo-style methods (with inner and outer optimizers). In particular, the literature suggests that compression + error feedback *maintains* performance while reducing communication to a point [1.1-1.4], and that Nesterov outer momentum is crucial for DiLoCo [1.9]. On the other hand, in this work, we show that *replacing* DiLoCo’s Nesterov outer momentum with a form of error feedback and a specific compressor applied on top of a local optimizer can actually improve performance (while still achieving the drastic communication improvements).  This allows us to beat the state of the art DiLoCo with Nesterov outer momentum. Our paper uses several empirical ablations and theoretical analysis to trace this surprising and novel view of topk+error feedback to a link with the outer momentum.

We summarize our replies to the reviewers’ weaknesses below:

---

> ### Author Response · Authors · 2025-12-03
>
> **AinH** | score: 4/10, confidence: 3/5 | Reviewer Reply: N/A
>
> ---
>
> **AinH Weaknesses:** (a) Theory is missing regarding local steps, (b) missing experimental setting for Table 3, and (c) fine-tuning experiments are missing.
>
> **Author’s reply to AinH:** We (a) provide new theory about SparseLoCo’s convergence in section I, (b) we note that the setting was included on L211-215 of the original manuscript but we now also include these details in Table 3’s caption, and (c) we emphasize that our experiments center around pre-training since communication efficient training algorithms like SparseLoCo are most useful when training large models over the internet and across datacenters, which is of most interest for pre-training but not fine-tuning. Our experimental focus on pre‑training also follows existing communication‑efficient baselines (DiLoCo, DeMo, StreamingDiLoCo), which are evaluated exclusively in the LLM pre‑training setting.
>
>
> &nbsp;
>
>
> **sx44** | score: 4/10, confidence: 2/5 | Reviewer Reply: N/A
>
> ---
>
> **sx44 Weaknesses:** (a) limited novelty beyond error feedback, (b) insufficient evidence of training speedup, and (c) our method is a generic application of sparsification.
>
> **Author’s reply to sx44:** (a) see the section above on **”Reviewer’s confusion about limited novelty”**; (b) we emphasize that SparseLoCo can support much lower bandwidths while achieving higher performance (as shown in Figure 2), directly implying wall clock time savings; and (c) we state that: “It is demonstrably false to state that the technique is a generic application of sparsification to DiLoCo. Previous works [2.1] and [2.7] have directly or generically applied sparsification to compress DiLoCo’s pseudogradient, resulting in a significant degradation in
> performance at low sparsities (see Figure 11 of [2.1] and Figure 2 of [2.7]). In contrast, we develop a subtle methodology explained in sections 3.1, 3.2, and 3.3 that allows us to aggressively sparsify the DiLoCo pseudogradient while improving the performance”.
>
>
> &nbsp;
>
> **aWwE** | score: 2/10, confidence: 3/5 | Reviewer Reply: N/A
>
> ---
>
> **aWwE Weaknesses:** (a) Novelty beyond DeMo is not clear from the paper, and (b) the writing could be more polished.
>
> **Author’s reply to aWwE:** (a) see the section above on **”Reviewer’s confusion about limited novelty”**; (b) we make improvements to the writing throughout the manuscript (highlighted in blue).
>
> &nbsp;
>
> **Ag2i** | score: 4/10, confidence: 3/5 | Reviewer Reply: N/A
>
> ---
>
> **Ag2i Weaknesses:** (a) limited novelty since error feedback and gradient compression exist, (b) reasons why the method works to outperform DiLoCo are not clear, and (c) needing to tune sparsity adds an extra hyperparameter.
>
> **Author’s reply to Ag2i:** (a) see the section above on **”Reviewer’s confusion about limited novelty”**; (b) we provide an explanation to the reviewer and have updated the manuscript accordingly in lines 351-365; (c) We note that many reasonable choices of sparsity still lead to more Pareto optimal solutions than existing SOTA baselines, while a properly tuned sparsity leads to stronger performance. Also, the optimal sparsity is consistent across the number of workers and model sizes, and seems to be only affected by the number of inner steps; therefore, additional tuning of sparsity is not vital for SparseLoCo’s performance.

---

### Meta-Review · Area_Chair_nogV · 2025-12-25

**Summary:**

The work studies communication-efficient distributed algorithm SparseLoCo for training LLMs in bandwidth-constrained environments. The proposed algorithm leverages error feedback with top-k and 2-bit quantization to reach low compression rates of 1-3% while outperforming full-precision training.

**Summary of the reviewers' concerns:**

1. Theoretical explanation or brief analytical justification (not necessarily a full convergence proof) is needed to clarify how the different components of the algorithm interact and jointly contribute to the observed behavior. (Reviewers AinH and aWwE)

2. Wider range of tasks and model families are needed to demonstrate generalizability and to support claims of broader applicability. (Reviewer AinH and sx44)

3. The core technique of error feedback under communication compression is an established mechanism for achieving communication efficiency. The paper demonstrates a limited algorithmic novelty relative to DeMo and DiLoCo. (Reviewers sx44, aWwE and Ag2i)

4. The paper lacks results demonstrating improved training efficiency in terms of training loss versus wall-clock time, which is necessary to substantiate claims of acceleration. (Reviewer sx44)

5. The presentation would benefit from improvements in clarity, organization, and narrative flow. (Reviewers AinH and aWwE)

**Reviewer Concerns:**

I think the rebuttal addressed concerns 4th and 5th above, but only partially addressed the remaining issues.

**Reviewer Scores:**

From my reading of the rebuttal, there might have been some score increases; however, I find the chances of this to be low. In particular, I beleive the following concerns still stand and keep the paper below the acceptance threshold:

- Limited empirical validation beyond pre-training on LLaMA-style architectures;
- Limited algorithmic novelty beyond what has already been proposed in DeMo and DiLoco;
- Insufficient theoretical justification, including a lack of proper discussion around the analysis.

I would also like to comment on the new convergence results added in Section I.

First, the error-feedback update in line 1043 does not seem to match the SparseLoCo update in line 235. There does not seem to exist any Cauchy–Young inequality mentioned in the proof. Furthermore, the result for LocalAdam appears to be a high-probability bound, whereas Section I claims an in-expectation bound. These require further clarification.

While the attempt to provide a convergence analysis is appreciated, the purpose of theoretical analysis goes beyond merely establishing analytical convergence. It should also help justify key aspects of the algorithmic design clarifying implications, relating the theory to practical design choices, and offering insights that either match or complement the empirical findings. At present, these connections remain insufficiently developed.

---

### Decision · Program_Chairs · 2026-01-26

Reject